# AN ONLINE LEARNING APPROACH TO PROMPT-BASED SELECTION OF GENERATIVE MODELS

## ABSTRACT

Selecting a sample generation scheme from multiple text-based generative models is typically addressed by choosing the model that maximizes an averaged evaluation score. However, this score-based selection overlooks the possibility that different models achieve the best generation performance for different types of text prompts. An online identification of the best generation model for various input prompts can reduce the costs associated with querying sub-optimal models. In this work, we explore the possibility of varying rankings of text-based generative models for different text prompts and propose an online learning framework to predict the best data generation model for a given input prompt. The proposed framework adapts the kernelized contextual bandit (CB) methodology to a CB setting with shared context variables across arms, utilizing the generated data to update a kernel-based function that predicts which model will achieve the highest score for unseen text prompts. Additionally, we apply random Fourier features (RFF) to the kernelized CB algorithm to accelerate the online learning process and establish a $\widetilde{\mathcal{O}}(\sqrt{T})$ regret bound for the proposed RFF-based CB algorithm over $T$ iterations. Our numerical experiments on real and simulated text-to-image and image-to-text generative models show RFF-UCB performs successfully in identifying the best generation model across different sample types.

## 1 INTRODUCTION

Text-based generative artificial intelligence (AI) has found numerous applications in various engineering tasks. A prompt-based generative AI represents a conditional generative model that produces samples given an input text prompt. Over the past few years, several frameworks using diffusion models and generative adversarial networks have been proposed to perform text-guided sample generation tasks for various data domains including image, audio, and video (Reed et al., 2016; Pan et al., 2018; Xu et al., 2018; Ding et al., 2021; Singer et al., 2022; Huang et al., 2023; Podell et al., 2024). The multiplicity of developed prompt-based models has led to significant interest in developing evaluation mechanisms to rank the existing models and find the best generation scheme. To address this task, several evaluation metrics have been proposed to quantify the fidelity and relevance of samples created by prompt-based generative models, such as CLIPScore (Hessel et al., 2021) and PickScore (Kirstain et al., 2023).

The existing model selection methodologies commonly aim to identify the generative model with the highest relevance score, producing samples that correlate the most with input text prompts. A well-known example is the CLIPScore for image generation models, measuring the expected alignment between the input text and output image of the model using the CLIP embedding (Radford et al., 2021b). While the best-model identification strategy has been frequently utilized in generative AI applications, this approach does not consider the possibility that the involved models can perform differently across text prompts. However, it is possible that one model outperforms another model in responding to text prompts from certain categories, while that model performs worse in generating samples for other text categories. Figure 1 shows one such example where two standard text-to-image models exhibit different rankings on text prompts with the terms "dog" and "car". In general, the different training sets and model architectures utilized to train text-based models can result in the models' varying performance in response to different text prompts, which is an important consideration in the selection of text-based generative models for various text prompts. We provide more examples in Figures 17 and 18 in the Appendix.

| | **Stable Diffusion** | **PixArt-$\alpha$** | **Examples** (clockwise) |
|---|---|---|---|

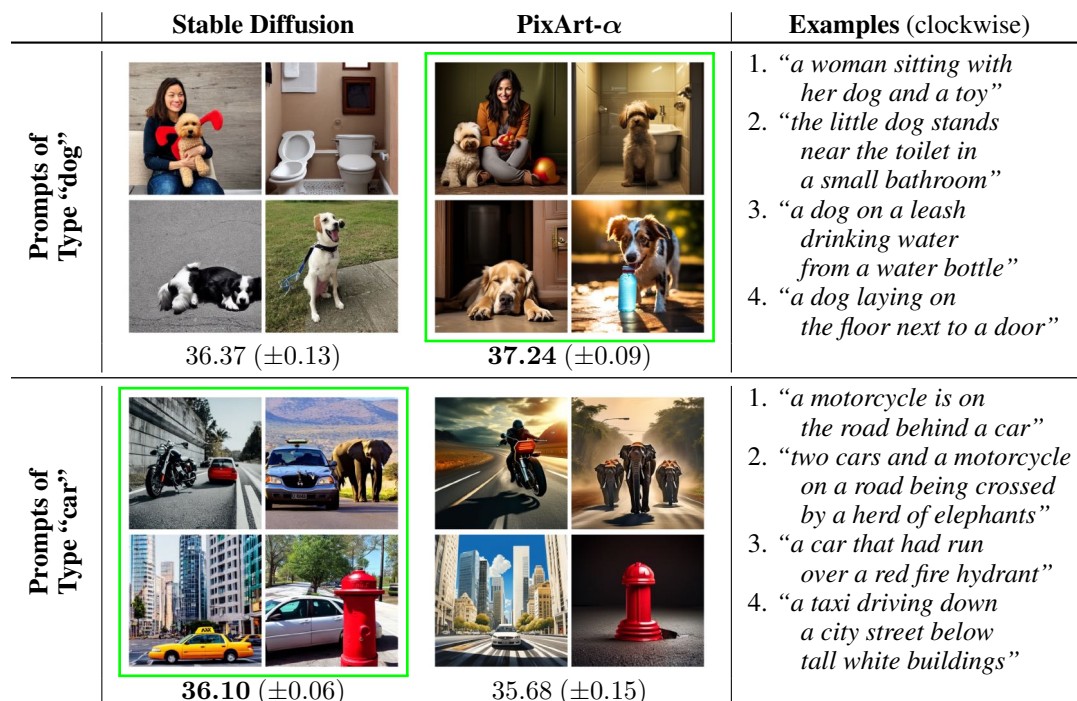

| | Stable Diffusion | PixArt-$\alpha$ | |
|---|---|---|---|
| **Prompts of Type "dog"** | 36.37 ($\pm$0.13) | **37.24** ($\pm$0.09) | 1. *"a woman sitting with her dog and a toy"* 2. *"the little dog stands near the toilet in a small bathroom"* 3. *"a dog on a leash drinking water from a water bottle"* 4. *"a dog laying on the floor next to a door"* |
| **Prompts of Type "car"** | **36.10** ($\pm$0.06) | 35.68 ($\pm$0.15) | 1. *"a motorcycle is on the road behind a car"* 2. *"two cars and a motorcycle on a road being crossed by a herd of elephants"* 3. *"a car that had run over a red fire hydrant"* 4. *"a taxi driving down a city street below tall white buildings"* |

Figure 1: Prompt-based generated images from Stable Diffusion and PixArt-$\alpha$: Stable Diffusion attains a higher CLIPScore in generating type-2 prompts (36.10 versus 35.68) while underperforms for type-1 prompts (36.37 versus 37.24).

In this work, we aim to develop a learning algorithm to identify the best generative model for a given input prompt, using observed prompt/generated samples collected from the models in the previous sample generation iterations. Since the goal of such text-based model selection is to minimize the data queries from suboptimal generative models for an input text prompt, we view the model selection task as an online learning problem, where after each data generation the learner updates a function predicting which generative model performs the best in response to different text prompts. Here, the goal of the online learner is to utilize the previously generated samples to accurately guess the generation model with the best performance for the incoming text prompt. An optimal online model selection method will result in a bounded regret value, measured in comparsion to the sample generation from the groundtruth-best model for the text prompts.

The described online learning task can be viewed as a *contextual bandit (CB)* problem studied in the multi-armed bandit literature (Langford & Zhang, 2007; Li et al., 2010). In a CB task, the online learner observes the context variable (the text prompt in our setting) and guesses the best arm for the current input context. Specifically, we focus on the kernelised upper confidence bound (kernel-UCB) approach and adapt this methodology to propose the *Shared-Context Kernel UCB (*SCK-UCB*)* for an online prompt-based selection of generative models. According to the SCK-UCB approach, the learner utilizes a UCB-score from a kernel-based prediction function to choose the generative model for the incoming text prompt and subsequently update the kernel-based prediction rule based on the generated data for the upcoming iterations. We prove that the proposed SCK-UCB achieves an $\widetilde{O}(\sqrt{T})$ regret bound over a horizon of $T$ iterations.

Since the user applying the CB-based model selection approach may have limited compute power and not be able to afford growing computational costs in the online learning process, we propose to utilize the random Fourier features (RFF) framework (Rahimi & Recht, 2007b) to balance the computational load between the iterations of SCK-UCB. We discuss that in the kernel-UCB methods, including our proposed SCK-UCB, the computational cost per iteration will grow cubically as $O(t^3)$ with iteration $t$. To address the growing cost per iteration of kernel-UCB, we leverage the RFF approach and develop the proxy RFF-UCB algorithm which approximates the solution to SCK-UCB, while the computational costs grow only linearly $O(t)$. We show that the regret bound for SCK-

UCB will approximately hold for RFF-UCB, and therefore, RFF-UCB provides an efficient proxy to the SCK-UCB algorithm, which can be run in devices with lower computation budget.

Finally, we present the results of several numerical experiments to show the efficacy of our proposed SCK-UCB and RFF-UCB in the online selection of conditional generative models. In our experiments, we simulate several text-to-image and image-captioning (image-to-text) models, where different models lead to different rankings of CLIPScore values across sample types. Our numerical results suggest a fast convergence of the proposed online learning algorithms to the best model available for different prompt types. Moreover, we apply the RFF-UCB method to several standard text-to-image models, and show how the algorithm can infer the model with higher CLIPScore with a growing accuracy as the iteration grows. In our experiments, the proposed SCK-UCB and RFF-UCB outperform the greedy baselines without any bonus term to encourage exploration in the learning process. The main contributions of our work can be summarized as:

- Studying the prompt-based selection of conditional generative models to improve the performance scores over every individual model

- Developing the contextual bandit-based SCK-UCB and RFF-UCB algorithms for the online selection of prompt-based generative models

- Providing the theoretical analysis of the regret and computational costs of SCK-UCB and RFF-UCB online learning methods

- Presenting numerical results on the online selection of generative models based on the incoming prompt using SCK-UCB and RFF-UCB

## 2    RELATED WORK

**(Automatic) Evaluation of conditional generative models.**    Evaluating the conditional generative models has been studied extensively in the literature. For text-to-image (T2I) generation, earlier methods primarily rely on the Inception score (Salimans et al., 2016) and Fréchet inception distance (Heusel et al., 2017). More recent works propose reference-free metrics for robust automatic evaluation of T2I and image captioning, with notable examples being CLIPScore (Hessel et al., 2021) and PickScore (Kirstain et al., 2023). Kim et al. (2022) propose a mutual-information-based metric, which attains consistency across benchmarks, sample parsimony, and robustness. To provide a holistic evaluation of T2I models, several works focus on multi-objective evaluation. Astolfi et al. (2024) propose to evaluate conditional image generation in terms of *prompt-sample consistency*, *sample diversity*, and *fidelity*. Kannen et al. (2024) introduce a framework to evaluate T2I models regarding *cultural awareness* and *cultural diversity*. Masrourisaadat et al. (2024) examine the performance of several T2I models in generating images such as human faces and groups and present a social bias analysis. Another line of study explores evaluation approaches using large language models (LLMs). Tan et al. (2024) develop LLM-based evaluation protocols that focus on the *faithfulness* and *text-image alignment*. Peng et al. (2024) introduce a GPT-based benchmark for evaluating personalized image generation. For evaluation of text-to-video (T2V) generation, Huang et al. (2024) introduce VBench as a comprehensive evaluation of T2V models in terms of *quality* and *consistencey*.

**(Kernelized) Contextual bandits.**    The contextual bandits (CB) is an efficient framework for online decision-making with context information (Langford & Zhang, 2007; Foster et al., 2018), which is widely adopted in domains such as recommendation system and online advertisement (Li et al., 2010). A key to its formulation is the relationship between the context (vector) and the expected reward. In linear CB, the reward is assumed to be linear to the context vector (Li et al., 2010; Chu et al., 2011). To incorporate non-linearity, Valko et al. (2013) propose kernelized CB, which assumes the rewards are linear-realizable in a reproducing kernel Hilbert space (RKHS). However, the proposed algorithm requires solving a kernel ridge regression per iteration, whose computation and required space have polynomial dependence on the number of iterations. To address this problem, a line of study leverages the assumption that the kernel matrix is often approximately low-rank and uses Nyström approximations (Calandriello et al., 2019; 2020; Zenati et al., 2022). [Recently, a line of study utilizes (contextual) bandit algorithms to improve the performance of generative models (Chen et al., 2024; Lin et al., 2024).]

# 3 PRELIMINARIES

## 3.1 CLIPSCORE

CLIPScore (Hessel et al., 2021) is a widely used automatic metric to evaluate the alignment of text-to-image/video (T2I/V) and image captioning models. Let $(y, x) \in \mathcal{Y} \times \mathcal{X}$ be any *text-image pair*. We denote by $\boldsymbol{c}_y \in \mathbb{S}^{d-1} := \{z \in \mathbb{R}^d : \|z\|_2 = 1\}$ and $\boldsymbol{v}_x \in \mathbb{S}^{d-1}$ the (normalized) embeddings of text $y \in \mathcal{Y}$ and image $x \in \mathcal{X}$, respectively, both extracted by CLIP (Radford et al., 2021a). The CLIPScore (Hessel et al., 2021) is given by

$$\text{CLIPScore}^{\text{T2I}}(y, x) := \max\{0, 100 \cdot \cos(\boldsymbol{v}_x, \boldsymbol{c}_y)\}, \tag{1}$$

and note that $\cos(\boldsymbol{v}_x, \boldsymbol{c}_y) = \langle \boldsymbol{v}_x, \boldsymbol{c}_y \rangle$ as we operate under the normalized embeddings. Further, for a video $X := \{x^{(l)}\}_{l=1}^L$ consisting of $L$ frames, where $x^{(l)}$ is the $l$-th frame, the score is the averaged frame CLIPScore, that is,

$$\text{CLIPScore}^{\text{T2V}}(y, X) := \frac{1}{L} \sum_{l=1}^L \text{CLIPScore}^{\text{T2I}}(y, x^{(l)}). \tag{2}$$

## 3.2 KERNEL METHODS AND RANDOM FOURIER FEATURES

Let $\phi : \mathbb{R}^d \to \mathcal{H}$ denote a mapping from the *primal space* $\mathbb{R}^d$ to the (possibly infinite-dimensional) associated *reproducing kernel Hilbert space* (RKHS) $\mathcal{H}$. The corresponding *kernel function* is defined by $k(y, y') := (\phi(y))^\top \phi(y')$ for any $y, y' \in \mathbb{R}^d$, where we use matrix notation $h_1^\top h_2 := \langle h_1, h_2 \rangle_{\mathcal{H}}$ to denote the inner product of two elements $h_1, h_2 \in \mathcal{H}$. The kernel function $k$ is *positive definite* if $\sum_{i=1}^n \sum_{j=1}^n c_i c_j k(y_i, y_j) \geq 0$ for any $n \in \mathbb{N}_+$, $y_1, \cdots, y_n \in \mathbb{R}^d$, and $c_1, \cdots, c_n \in \mathbb{R}$. In other words, the *kernel matrix* $K := [k(y_i, y_j)]_{i,j=1}^n \in \mathbb{R}^{n \times n}$ is always positive semi-definite (PSD). Further, a positive definite kernel function is *shift invariant* if $k(y, y') := k(y - y')$ for any $y, y' \in \mathbb{R}^d$. An example is the *radial basis function* (RBF) kernel, i.e., $k_{\text{RBF}}(y, y') = \exp(-\|y - y'\|_2^2/(2\sigma^2))$ with $\sigma > 0$.

**Kernel ridge regression (KRR).** [Given empirical data $(y_1, s_1), \cdots, (y_n, s_n)$, where $\{y_i \in \mathbb{R}^d\}_{i=1}^n$ are *dependent variables* and $\{s_i \in \mathbb{R}\}_{i=1}^n$ are *target variables*, respectively], the kernel method assumes the existence of $w^\star \in \mathcal{H}$ such that $\mathbb{E}[s_i|y_i] = (\phi(y))^\top w^\star$ for any $i = 1, \cdots, n$. Let $\alpha \geq 0$ denote a *regularization parameter*. KRR constructs the estimator $\widehat{s}_{\text{KRR}}(y) := k_y^\top (K + \alpha I_n)^{-1} v$ for any $y \in \mathbb{R}^d$, where $K = [k(y_i, y_j)]_{i,j=1}^n \in \mathbb{R}^{n \times n}$ is the kernel matrix, $v := [s_1, \cdots, s_n]^\top \in \mathbb{R}^n$, and $k_y = [k(y_1, y), \cdots, k(y_n, y)]^\top \in \mathbb{R}^n$. The estimator can be interpreted as first estimating $w^\star$ by ridge regression $\widehat{w} := \arg\min_{w \in \mathcal{H}} \sum_{i=1}^n ((\phi(y_i))^\top w - s_i)^2 + \alpha \|w\|$, where $\|w\| := \sqrt{w^\top w}$ for any $w \in \mathcal{H}$, and then making the prediction $\widehat{s}_{\text{KRR}}(y) = (\phi(y))^\top \widehat{w}$.

**Random Fourier features (RFF).** One problem of KRR is that it scales poorly with the size $n$ of the empirical data, i.e., computing the KRR estimator generally requires $\Omega(n^3)$ time and $\Omega(n^2)$ memory. To address this problem, Rahimi & Recht (2007a) propose to scale up kernel methods by RFF sampling. Specifically, the Bochner's Theorem (Rudin, 2017) implies that for any (properly scaled) shift-invariant kernel $k(y, y') = k(y - y')$, there exists a distribution $p \in \Delta(\mathbb{R}^d)$ such that $k(y, y') = \mathbb{E}_{w \sim p}[e^{iw^\top(y - y')}]$, where $e^{i\theta} := \cos\theta + i \cdot \sin\theta$ for any $\theta \in \mathbb{R}$ and $i$ is the *imaginary unit*. Therefore, the idea of RFF is to sample $w_1, \cdots, w_s \sim p$ and approximate $k(y, y')$ by the empirical mean $s^{-1} \sum_{j=1}^s e^{iw_j^\top(y - y')}$ to within $\epsilon$ with only $s = O(d\epsilon^{-2} \log(1/\epsilon^2))$. Since the kernel $k(\cdot)$ is real, we can replace the complex exponentials with cosines and define

$$\varphi(y) := \sqrt{\frac{2}{s}} \cdot [\cos(w_j^\top y + b_j), \cdots, \cos(w_s^\top y + b_s)]^\top, \text{ where } w_j \overset{\text{i.i.d}}{\sim} p, b_j \overset{\text{i.i.d}}{\sim} \text{Unif}([0, 2\pi]) \tag{3}$$

and $k(y, y')$ is approximated by $(\varphi(y))^\top \varphi(y')$. The resulting approximate KRR estimator $\widetilde{s}_{\text{KRR}}(y) := (\widetilde{\Phi}^* \widetilde{\Phi} + \alpha I_s)^{-1} \widetilde{\Phi}^* v$, where $\Phi := [\varphi(y_i)^\top]_{i=1}^n \in \mathbb{C}^{n \times s}$, can be computed in $O(ns^2)$ time and $O(ns)$ memory, giving substantial computational savings if $s \ll n$ (Avron et al., 2017). For the RBF kernel, the distribution $p_{\text{RBF}}$ is the multivariate Gaussian $\mathcal{N}(0, \sigma^{-2} \cdot I_d)$.

## 4 PROMPT-BASED SELECTION AS CONTEXTUAL BANDITS

In this section, we introduce the framework of online prompt-based selection of generative models, which is given in Protocol 6. Let $[N] := \{1, \cdots, N\}$ for any positive integer $N \in \mathbb{N}_+$. We denote by $\mathcal{G} := [G]$ the set of (prompt-based) generative models. The evaluation proceeds in $T \in \mathbb{N}_+$ iterations. At any iteration $t \in [T]$, a *prompt* $y_t \in \mathcal{Y}$ is drawn from a fixed distribution $\rho \in \Delta(\mathcal{Y})$ on the prompt space $\mathcal{Y} \subseteq \mathbb{S}^{d-1}$, e.g., (the normalized embedding of) a picture in image captioning or a paragraph in text-to-image/video generation. Based on prompt $y_t$ (and previous observation sequence), an algorithm $\mathcal{A}$ picks model $g_t \in \mathcal{G}$ and samples an *answer* $x_t \sim P_{g_t}(\cdot|y_t)$, where $P_g(\cdot|y) \in \Delta(\mathcal{X})$ is the conditional distribution of answers generated from any model $g \in \mathcal{G}$. The quality of answer $x_t$ is given by $s(y_t, x_t)$, where $s : \mathcal{Y} \times \mathcal{X} \to [-1, 1]$ is the *score function*. The algorithm $\mathcal{A}$ aims to minimize the *regret*

$$\text{Regret}(T) := \sum_{t=1}^{T} \left( s_\star(y_t) - s_{g_t}(y_t) \right), \tag{4}$$

where we denote by $s_g(y) := \mathbb{E}_{x_g \sim P_g(\cdot|y)}[s(y, x_g)]$ the expected score of any model $g \in \mathcal{G}$ and $s_\star(y) := \max_{g \in \mathcal{G}} s_g(y)$ the optimal expected score, both conditioned to prompt $y$.

---

**Protocol 1** Online Prompt-based Selection of Generative Models

---

**Require:** total iterations $T \in \mathbb{N}_+$, set of generators $\mathcal{G} = [G]$, prompt distribution $\rho \in \Delta(\mathcal{Y})$, score function $s : \mathcal{Y} \times \mathcal{X} \to [-1, 1]$, algorithm $\mathcal{A} : (\mathcal{Y} \times \mathcal{G} \times \mathbb{R})^* \times \mathcal{Y} \to \Delta(\mathcal{G})$
**Initialize:** observation sequence $\mathcal{D} \leftarrow \varnothing$
1: **for** iteration $t = 1, 2, \cdots, T$ **do**
2:      Prompt $y_t \sim \rho$ is revealed.
3:      Algorithm $\mathcal{A}$ picks model $g_t \sim \mathcal{A}(\cdot|\mathcal{D}, y_t)$ and samples an answer $x_t \sim P_{g_t}(\cdot|y_t)$.
4:      Score $s_t \leftarrow s(y_t, x_t)$ is assigned.
5:      Update observation sequence $\mathcal{D} \leftarrow \mathcal{D} \cup \{(y_t, g_t, s_t)\}$.
6: **end for**

---

## 5 AN OPTIMISM-BASED APPROACH FOR PROMPT-BASED SELECTION

Under the setting of online prompt-based selection, a key challenge is to learn the relationship between the prompt and the expected score for each model. In this paper, we assume the scores are linear to the prompt vector in the reproducing kernel Hilbert space (RKHS) with model-dependent weights.

**Assumption 1** (Realizability). *There exists a mapping $\phi : \mathbb{R}^d \to \mathcal{H}$ and weight $w_g^\star \in \mathcal{H}$ such that score $s_g(y) = \langle y, w_g^\star \rangle_\mathcal{H}$ for any prompt vector $y \in \mathbb{R}^d$ and model $g \in \mathcal{G}$. Further, it holds that $\|w_g^\star\| \leq 1$, and $k(y, y) \leq \kappa^2$ and $\|\phi(y)\| \leq 1$ for any $y \in \mathcal{Y}$, where $k : \mathbb{R}^d \times \mathbb{R}^d \to \mathbb{R}$ is the kernel function of the mapping $\phi$.*

**Remark 1** (Shared context with kernelized rewards). *We note that Assumption 1 is slightly different from the one made in kernelized bandits (Valko et al., 2013; Zenati et al., 2022), where a context is observed per each arm and assumes the existence of a shared weight. [However, in the prompt-based generation setting, there is a single prompt at each iteration and responses can vary across the models, which leads to our formulation of shared context and model-dependent weights.]*

**Remark 2.** *Our assumption of linear-realizable scores in RKHS is motivated by the following observations. First, the relationship between the prompt vector and score is often highly non-linear and generator-dependent. For instance, the generated images often vary across the prompts and different T2I models, which can have substantial effect on the resulting CLIPScore (1). Second, the kernel methods can approximate any function arbitrarily well with enough training data and enjoy nice statistical properties.*

### 5.1 THE SCK-UCB ALGORITHM

In this section, we present SCK-UCB in Algorithm 2, an optimism-based approach to the online evaluation of prompt-base generation. At each iteration, SCK-UCB first estimates the scores via

kernel ridge regression (KRR) and then picks the model with the highest estimated score. To construct the KRR dataset, the algorithm maintains index sets $\{\Psi_g\}_{g \in \mathcal{G}}$, where each set $\Phi_g \subseteq [T]$ stores the iterations such that model $g$ is chosen (line 6).

---

**Algorithm 2** Shared-Context Kernel UCB (SCK-UCB)

---

**Require:** total iterations $T \in \mathbb{N}_+$, set of generators $\mathcal{G} = [G]$, prompt distribution $\rho \in \Delta(\mathcal{Y})$, score function $s : \mathcal{Y} \times \mathcal{X} \to [-1, 1]$, positive definite kernel $k : \mathcal{Y} \times \mathcal{Y} \to \mathbb{R}$, regularization and exploration parameters $\alpha, \eta \geq 0$

**Initialize:** observation sequence $\mathcal{D} \leftarrow \varnothing$ and index set $\Psi_g \leftarrow \varnothing$ for all $g \in \mathcal{G}$

1: **for** iteration $t = 1, 2, \cdots, T$ **do**
2:      Prompt $y_t \sim \rho$ is revealed.
3:      Compute $\{(\widehat{\mu}_g, \widehat{\sigma}_g) \leftarrow \text{COMPUTE\_UCB}(\mathcal{D}, y_t, \Psi_g)\}_{g \in \mathcal{G}}$.
4:      Pick model $g_t \leftarrow \arg\max_{g \in \mathcal{G}}\{\widehat{s}_g\}$, where $\widehat{s}_g \leftarrow \widehat{\mu}_g + (2\eta + \sqrt{\alpha}) \cdot \widehat{\sigma}_g$.
5:      Sample an answer $x_t \sim P_{g_t}(\cdot|y_t)$ and compute the score $s_t \leftarrow s(y_t, x_t)$.
6:      Update $\mathcal{D} \leftarrow \mathcal{D} \cup \{(y_t, s_t)\}$ and $\Psi_{g_t} \leftarrow \Psi_{g_t} \cup \{t\}$.
7: **end for**

8: **function** COMPUTE\_UCB($\mathcal{D}, y, \Psi_g$)
9:      **if** $\Psi_g$ is empty **then**
10:        $\widehat{\mu}_g \leftarrow +\infty, \widehat{\sigma}_g \leftarrow +\infty$.
11:      **else**
12:        Set $K \leftarrow [k(y_i, y_j)]_{i,j \in \Psi_g}$, $v \leftarrow [s_i]_{i \in \Psi_g}^\top$, and $k_y \leftarrow [k(y, y_i)]_{i \in \Psi_g}^\top$.
13:        $\widehat{\mu}_g \leftarrow k_y^\top (K + \alpha I)^{-1} v$.
14:        $\widehat{\sigma}_g \leftarrow \alpha^{-\frac{1}{2}} \sqrt{k(y, y) - k_y^\top (K + \alpha I)^{-1} k_y}$.
15:      **end if**
16:      **return** $(\widehat{\mu}_g, \widehat{\sigma}_g)$.
17: **end function**

---

The key design in SCK-UCB is the function COMPUTE\_UCB (lines 8-17), which outputs both the KRR estimator $\widehat{\mu}_g$ and an uncertainty quantifier $\widehat{\sigma}_g$. The estimated score is then computed by $\widehat{s}_g = \widehat{\mu}_g + (2\eta + \sqrt{\alpha})\widehat{\sigma}_g$ (line 4), which is initially set to $+\infty$ to ensure each model is picked at least once (lines 9-10). Particularly, the following lemma shows that under some conditions, $\widehat{s}_g$ is an optimistic estimation of $s_g(y_t)$ with high probability. The detailed proof can be found in Appendix C.1.

**Lemma 1** (Optimism). *Let $\Psi_g \subseteq [T]$ be an index set such that the set of scores $\{s_t : t \in \Psi_g\}$ are independent random variables. Then, under Assumption 1, with probability at least $1 - \delta$, the quantity $\widehat{\mu}_g$ computed in function COMPUTE\_UCB($\mathcal{D}, y, \Psi_g$) satisfies that*

$$|\widehat{\mu}_g - s_g(y)| \leq (2\eta + \sqrt{\alpha})\widehat{\sigma}_g, \tag{5}$$

*where $\eta = \sqrt{2\log(2/\delta)}$. Hence, it holds that $\widehat{s}_g = \widehat{\mu}_g + (2\eta + \sqrt{\alpha})\widehat{\sigma}_g \geq s_g(y)$.*

We show that a variant of SCK-UCB attains a regret of $\widetilde{O}(\sqrt{GT})$. The formal statement and the proof can be found in Appendix A.

**Theorem 1** (Regret, informal). *Under the same conditions in Lemma 1, with probability of at least $1 - \delta$, a variant of Algorithm 2 attains a regret of $\widetilde{O}(\sqrt{GT})$.*

### 5.2 SCK-UCB WITH RANDOM FOURIER FEATURES

The SCK-UCB solves a KRR for each model at an iteration to estimate the scores, which can be expensive in both computation and memory for a large number of iterations. To address this problem, we leverage the random Fourier features (RFF) sampling (Rahimi & Recht, 2007a) for positive definite shift-invariant kernels. At a high level, RFF maps the input data, e.g., the prompt (vector) in our setting, to a randomized low-dimensional feature space and then applies fast linear methods to solve the regression problem. Particularly, the inner product between these projected randomized features is an *unbiased* estimation of the kernel value.

We present the RFF-UCB algorithm, which is a variant of SCK-UCB with random features. RFF-UCB leverages a RFF-based approach to compute the mean and uncertainty quantifier in line 3 of Algorithm 2, which we present in Algorithm 3. Upon receiving the regression dataset consisting of prompt-score pairs, COMPUTE_UCB_RFF first projects each $d$-dimensional prompt vector to a randomized $s$-dimensional feature space according to Equation (3) (lines 5-6) and then solves a linear ridge regression to estimate the mean and uncertainty (lines 8-9). To derive statistical guarantees, the number of features varies according to both the input data and error thresholds, which we will specify in Appendix B.1. In practice, we find that a size around 50 can attain satisfactory empirical performance. To see why RFF can reduce the computation, note that the size of the (regularized) Gram matrix $(\widetilde{\Phi}_g^\top \widetilde{\Phi}_g + \alpha I)$ in line 8 is fixed to be $s$ in the whole process, while the size of $(K + \alpha I)$ in line 13 of Algorithm 2 scales with $|\Psi_g|$ and can grow linearly over iterations. Particularly, the following lemma shows that COMPUTE_UCB_RFF can reduce the time and space by an order of $O(t^2)$ and $O(t)$, respectively.

---

**Algorithm 3** Compute UCB with Random Fourier Features

---

**Require:** the Fourier transform $p$ of a positive definite shift-invariant kernel $k(y, y') = k(y - y')$, error thresholds $\epsilon_{\text{RFF}}, \Delta_{\text{RFF}} > 0$, regularization and exploration parameters $\alpha, \eta \geq 0$
**Initialize:** number of features $s$, bonus terms $\mathcal{B}_{g,1}$ and $\mathcal{B}_{g,2}$
 1: **function** COMPUTE_UCB_RFF$(\mathcal{D}, y, \Psi_g)$
 2:      **if** $\Psi_g$ is empty **then**
 3:          $\widetilde{\mu}_g \leftarrow +\infty, \widetilde{\sigma}_g \leftarrow +\infty.$
 4:      **else**
 5:          Draw $\omega_1, \cdots, \omega_s \overset{\text{i.i.d.}}{\sim} p$ and $b_1, \cdots, b_s \overset{\text{i.i.d.}}{\sim} \text{Unif}([0, 2\pi]).$
 6:          Define mapping $\varphi(y') \leftarrow \sqrt{\frac{2}{s}} \cdot [\cos(w_1^\top y + b_1), \cdots, \cos(w_s^\top y + b_s)]^\top$ for any $y' \in \mathbb{R}^d.$
 7:          Set $\widetilde{\Phi}_g \leftarrow [\varphi(y_i)^\top]_{i \in \Psi_g}$ and $v \leftarrow [s_i]_{i \in \Psi_g}^\top.$
 8:          $\widetilde{\mu}_g \leftarrow (\varphi(y))^\top (\widetilde{\Phi}_g^\top \widetilde{\Phi}_g + \alpha I)^{-1} \widetilde{\Phi}_g^\top v + \mathcal{B}_{g,1}.$
 9:          $\widetilde{\sigma}_g \leftarrow \alpha^{-\frac{1}{2}} \sqrt{1 - (\varphi(y))^\top (\widetilde{\Phi}_g^\top \widetilde{\Phi}_g + \alpha I)^{-1} \widetilde{\Phi}_g^\top \widetilde{\Phi}_g (\varphi(y))} + \mathcal{B}_{g,2}.$
10:      **end if**
11:      **return** $(\widetilde{\mu}_g, \widetilde{\sigma}_g).$
12: **end function**

---

**Lemma 2** (Time and space complexity). *At any iteration $t \in [T]$, COMPUTE_UCB (lines 8-17 of Algorithm 2) requires $O(t^3/G^2)$ time and $O(t^2/G)$ space, while COMPUTE_UCB_RFF with random features of size $s \in \mathbb{N}_+$ (Algorithm 3) requires $O(ts^2)$ time and $O(ts)$ space, where $G$ is the number of generators. See Appendix B.5 for details.*

It can be shown that the implementation of SCK-UCB with RFF attains the exact same regret guarantees for adaptively selected feature sizes. The formal statement and the proof can be found in Appendix B.2.

**Theorem 2** (Regret when using RFF, informal). *Under the same conditions in Theorem 4, a variant of RFF-UCB attains a regret of $\widetilde{O}(\sqrt{GT})$.*

## 6 NUMERICAL RESULTS

In this section, we provide numerical results for the proposed SCK-UCB-poly3 algorithm (SCK-UCB using polynomial kernel with degree 3, i.e., $k_{\text{poly3}}(x_1, x_2) = (1 + x_1^\top x_2)^3$) and the RFF-UCB algorithm using RBF kernel on various prompt-based generation tasks, including text-to-image (T2I) generation, image captioning (image-to-text), and text-to-video (T2V) generation.

[**Baselines.** We compare the proposed methods with five baselines, including 1) Lin-UCB: SCK-UCB with linear kernel, i.e., $k_{\text{lin}}(x_1, x_2) = x_1^\top x_2$, which does not incorporate non-linearity in score estimation, 2) One-arm Oracle: always picking the model with the maximum averaged CLIPScore, 3) Naive-KRR: SCK-UCB-poly3 without exploration, which selects the model with the highest estimated mean conditioned to the prompt, 4) Greedy: always generating samples from the model with the highest empirical CLIPScore, and 5) Random: uniformly selecting a generator at each step. The results for baselines are presented by dot lines with different colors.]

**Performance metrics.** For each experiment, we report three performance metrics: [(i) *outscore-the-best* (O2B): the difference between the CLIPScore attained by the algorithm and the highest average CLIPScore attained by any single model], (ii) *optimal-pick-ratio* (OPR): the overall ratio that the algorithm picks the best generator conditioned to the prompt type, (iii) *moving-average OPR*: OPR over the last 100 iterations.

[**Summary of results.** The main finding of our numerical experiments is the improvement of the proposed contextual bandit SCK-UCB algorithm over the one-arm oracle baseline. This result means that the online learning algorithm can outperform a user with side-knowledge of the single best-performing model, which is made possible by a *prompt-based selection* of the model. This finding supports the application of contextual bandit algorithms in the selection of text-based generative models. Moreover, our numerical results indicate that the proposed SCK-UCB algorithm can perform better with a non-linear kernel function. Finally, in our experiments, the proposed RFF-UCB variant could reduce the computational costs of the general SCK-UCB algorithm.]

### 6.1 TEXT-TO-IMAGE GENERATION

**Setup 1: Prompt-based selection between real generative models.** The first set of experiments are on the setup illustrated in Figure 1, where we generate images from two T2I generators, including Stable Diffusion v1-5[1] and PixArt-$\alpha$-XL-2-512x512[2] (see Figure 2). The results show that SCK-UCB-poly3 outperforms the baseline algorithm and attains a high optimal-pick-ratio, which shows that it can identify the optimal model conditioned to the prompt. Additionally, we provide numerical results on various T2I generative models, including uni-Diffuser[3] and DeepFloyd IF-I-M-v1.0.[4] (see Figures 14, 15, and 16 in the Appendix).

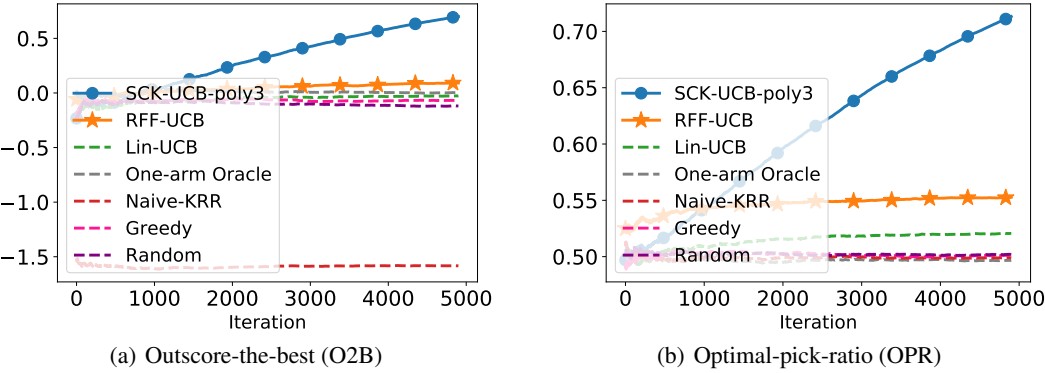

(a) Outscore-the-best (O2B)                    (b) Optimal-pick-ratio (OPR)

Figure 2: Prompt-based selection between Stable Diffusion and PixArt-$\alpha$ (Setup 1): Results are averaged over 20 trials.

[**Setup 2: Adapt to newly-introduced prompts and generators.** We consider scenarios where new generative models or prompt types are introduced after the initial deployment. In the first experiment, there are two available generators initially, including Stable Diffusion and PixArt-$\alpha$. After 2,500 iterations, uniDiffuser is also available (see Figure 3). In the second experiment, we generate samples from both PixArt-$\alpha$ and uniDiffuser, and a new prompt type is introduced after each 1,000 iterations (see Figure 13 in the Appendix). The results show that SCK-UCB-poly3 can well adapt to new prompt types and generators.]

**Setup 3: Synthetic expert T2I models.** In this setup, we synthesize five T2I generators based on Stable Diffusion 2, where each generator is an "expert" in generating images corresponding to a prompt type. The prompts are captions in the MS-COCO dataset from five categories: dog, car, carrot, cake, and bowl. At each iteration, a caption is drawn from a (random) category, and an image

---

[1]https://huggingface.co/docs/diffusers/en/api/pipelines/stable_diffusion/text2img

[2]https://huggingface.co/PixArt-alpha/PixArt-XL-2-512x512

[3]https://github.com/thu-ml/unidiffuser

[4]https://github.com/deep-floyd/IF

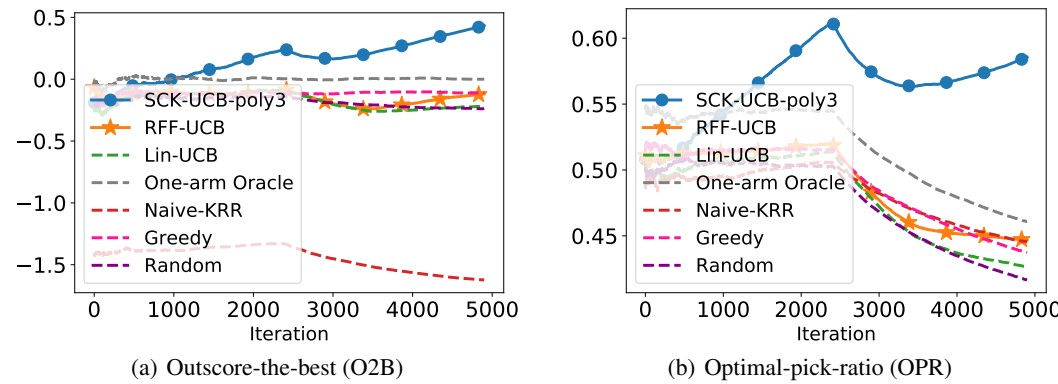

(a) Outscore-the-best (O2B)      (b) Optimal-pick-ratio (OPR)

Figure 3: Adapt to newly-introduced generators (Setup 2): Results are averaged over 20 trials.

is generated from Stable Diffusion 2. If the learner does not select the expert generator, then we add Gaussian noise to the generated image. Examples are visualized in Figure 4.

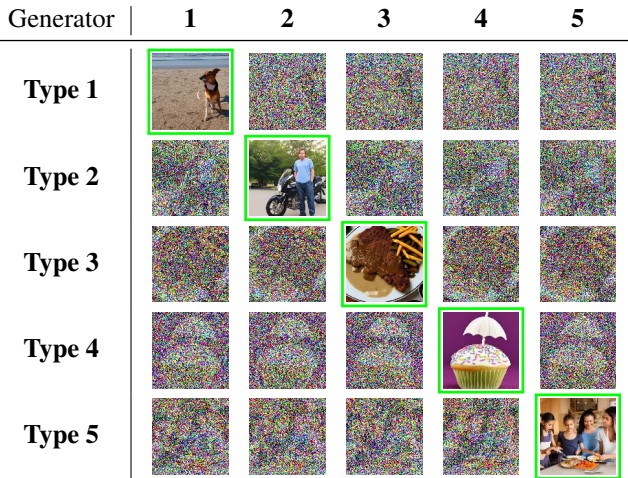

Figure 4: Generated images with noise perturbations: Each row and column display the generated images from a synthetic generator according to one single type of prompts. Images generated by the expert models are framed by green boxes. Gaussian noises are applied to non-expert models.

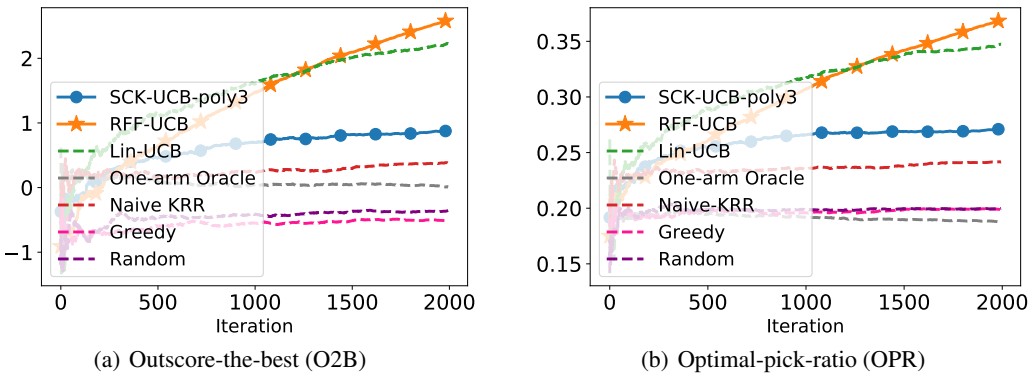

(a) Outscore-the-best (O2B)      (b) Optimal-pick-ratio (OPR)

Figure 5: Synthetic expert T2I models (Setup 3): Results are averaged over 20 trials.

## 6.2 RESULTS ON OTHER PROMPT-BASED GENERATION TASKS

**Setup 4: Image Captioning.** In this setup, the images are chosen from the MS-COCO dataset from five categories: dog, car, carrot, cake, and bowl. Similar to Section 6.1, we synthesize five expert generators based on vit-gpt2 model in the Transformers repository.[5] If a non-expert generator is chosen, then the caption is generated from the noisy image perturbed by Gaussian noises. Examples are visualized in Figure 19. The numerical results are summarized in Figure 6.

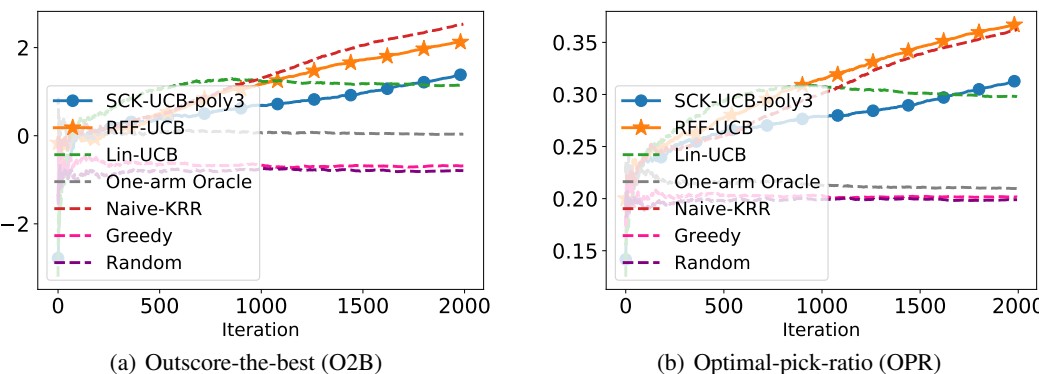

(a) Outscore-the-best (O2B)    (b) Optimal-pick-ratio (OPR)

Figure 6: Image captioning (Setup 4): Results are averaged over 20 trials.

**Setup 5: Synthetic Text-to-Video (T2V) task.** We provide numerical results on a synthetic T2V setting. Specifically, both the captions and videos are randomly selected from the following five categories of the MSR-VTT dataset (Xu et al., 2016): sports/action, movie/comedy, vehicles/autos, music, and food/drink. Each of the five synthetic arms corresponds to an expert in "generating" videos from a single category. Gaussian noises are applied to the video for non-experts. The results are summarized in Figure 7.

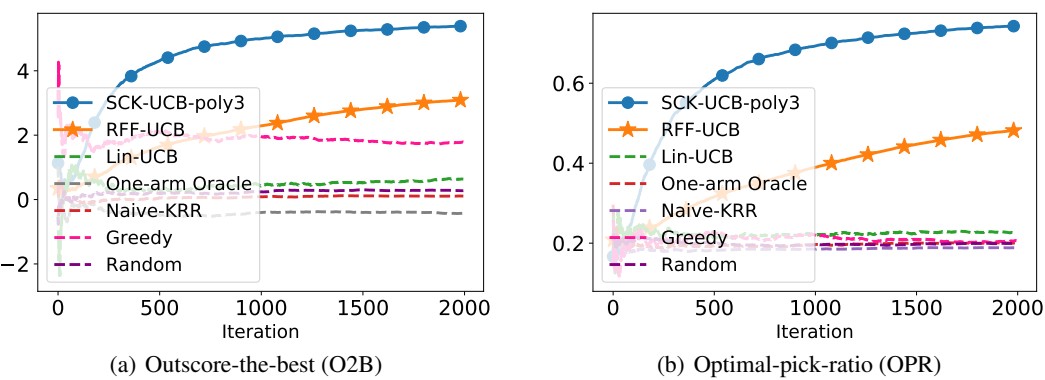

(a) Outscore-the-best (O2B)    (b) Optimal-pick-ratio (OPR)

Figure 7: Synthetic T2V task (Setup 5): Results are averaged over 20 trials.

## 7 CONCLUSION

In this work, we investigated prompt-based selection of generative models using a contextual bandit algorithm, which can identify the best available generative model for a given text prompt. We adapted the Kernel-UCB algorithm to perform this selection task and proposed two new algorithms: SCK-UCB and RFF-UCB. Our numerical results on text-to-image, text-to-video, and image-captioning tasks demonstrate the effectiveness of the proposed framework in scenarios where the available generative models have varying performance rankings depending on the type of prompt.

---

[5]https://huggingface.co/nlpconnect/vit-gpt2-image-captioning

An interesting direction for future research is to extend the application of our algorithms to text-to-text language models, where different models may respond better to questions on different topics. Furthermore, considering evaluation criteria beyond relevance, such as diversity and novelty scores, could lead to extensions of our proposed framework.

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

# A  PROOF IN SECTION 5.1

The technical challenge in analyzing SCK-UCB is that predictions in later iterations are made use of previous outcomes. Hence, the rewards $\{s_i\}_{i \in \Psi_g}$ are not independent if the index set $\Phi_g$ is updated each time when model $g$ is chosen (line 6 of Algorithm 2). To address this problem, we leverage a standard approach used in prior works (Auer, 2003; Chu et al., 2011; Valko et al., 2013) and present a variant of SCK-UCB in Algorithm 4, which is called Sup-SCK-UCB. We prove it attains a regret of order $\widetilde{O}(\sqrt{T})$.

## A.1  THE Sup-SCK-UCB ALGORITHM

---
**Algorithm 4** Sup-SCK-UCB
---
**Require:** total iterations $T \in \mathbb{N}_+$, set of generators $\mathcal{G} = [G]$, prompt distribution $\rho \in \Delta(\mathcal{Y})$, score function $s : \mathcal{Y} \times \mathcal{X} \to [-1, 1]$, positive definite kernel $k : \mathcal{Y} \times \mathcal{Y} \to \mathbb{R}$, regularization and exploration parameters $\alpha, \eta \geq 0$, function COMPUTE_UCB in Algorithm 2
**Initialize:** observation sequence $\mathcal{D} \leftarrow \varnothing$ and index sets $\{\Psi_g^m \leftarrow \varnothing\}_{m=1}^M$ for all $g \in \mathcal{G}$, where $M \leftarrow \log T$
 1: **for** iteration $t = 1, 2, \cdots, T$ **do**
 2:      Prompt $y_t \sim \rho$ is revealed.
 3:      Set stage $m \leftarrow 1$ and $\widehat{\mathcal{G}}^1 \leftarrow \mathcal{G}$.
 4:      **repeat**
 5:          Compute $\{(\widehat{\mu}_g^m, \widehat{\sigma}_g^m) \leftarrow \text{COMPUTE\_UCB}(\mathcal{D}, y_t, \Psi_g^m)\}_{g \in \widehat{\mathcal{G}}^m}$.
 6:          Set upper confidence bound $\widehat{s}_g^m(y) \leftarrow \widehat{\mu}_g^m + (2\eta + \sqrt{\alpha}) \cdot \widehat{\sigma}_g^m$ for all $g \in \widehat{\mathcal{G}}^m$.
 7:          **if** $(2\eta + \sqrt{\alpha}) \cdot \widehat{\sigma}_g^m \leq 1/\sqrt{T}$ for all $g \in \widehat{\mathcal{G}}^m$ **then**
 8:              Pick model $g_t \leftarrow \arg\max \widehat{s}_g^m(y)$.
 9:          **else if** $(2\eta + \sqrt{\alpha}) \cdot \widehat{\sigma}_g^m \leq 2^{-m}$ for all $g \in \widehat{\mathcal{G}}^m$ **then**
10:              $\widehat{\mathcal{G}}^{m+1} \leftarrow \{g \in \widehat{\mathcal{G}}^m : \widehat{s}_g^m(y) \geq \max_{g \in \widehat{\mathcal{G}}^m} \widehat{s}_g^m(y) - 2^{1-m}\}$.
11:              Set stage $m \leftarrow m + 1$.
12:          **else**
13:              Pick $g_t \in \widehat{\mathcal{G}}^m$ such that $(2\eta + \sqrt{\alpha}) \cdot \widehat{\sigma}_g^m > 2^{-m}$.
14:              Update $\Psi_{g_t}^m \leftarrow \Psi_{g_t}^m \cup \{t\}$.
15:          **end if**
16:      **until** a model $g_t$ is selected
17:      Sample an answer $x_t \sim P_{g_t}(\cdot|y_t)$ and compute the score $s_t \leftarrow s(y_t, x_t)$.
18:      Update $\mathcal{D} \leftarrow \mathcal{D} \cup \{(y_t, s_t)\}$.
19: **end for**
---

## A.2  ANALYSIS

In this section, we prove the following regret bound of Sup-SCK-UCB.

**Theorem 3** (Regret of Sup-SCK-UCB)**.** *Under Assumption 1, with probability at least $1 - \delta$, the regret of* Sup-SCK-UCB *with $\eta = \sqrt{2 \log(2(\log T) G T / \delta)}$ is bounded by*

$$\text{Regret}(T) \leq \widetilde{O}\left((1 + \sqrt{\alpha}) \sqrt{d_{\text{eff}}\left(10 + \frac{15}{\alpha}\right) GT}\right) \tag{6}$$

*where $d_{\text{eff}}$ is a data-dependent quantity defined in Lemma 10 and logarithmic factors are hidden in the notation $\widetilde{O}(\cdot)$.*

**Notations.** To facilitate the analysis, we add an subscript $t$ to all the notations in Algorithm 4 to indicate they are quantities computed at the $t$-th iteration, i.e., $\widehat{\mu}_{g,t}^m, \widehat{\sigma}_{g,t}^m, \widehat{s}_{g,t}^m, \widehat{\mathcal{G}}_t^m$, and $\Psi_{g,t}^m$.[6]

---

[6]Note that line 14 in the algorithm is rewritten as "$\Psi_{g_t,t+1}^m \leftarrow \Psi_{g_t,t+1}^m \cup \{t\}, \Psi_{g_t,t+1}^{m'} \leftarrow \Psi_{g_t,t}^{m'}$ for any $m' \neq m$, and $\Psi_{g,t+1}^{m'} \leftarrow \Psi_{g,t}^{m'}$ for any $g \neq g_t$ and $m \in [M]$". In addition, we set $\Psi_{g,t+1}^m \leftarrow \Psi_{g,t+1}^m$ for all $g \in \mathcal{G}$ and $m \in [M]$ in line 8.

We leverage the following two lemmas to prove Theorem 3. The first lemma shows that the construction of index sets $\{\Psi_{g,t}^m\}_{m=1}^M$ ensures the independence among the rewards $\{s_i\}_{i \in \Psi_{g,t}^m}$, which allows us to utilize Lemma 1 to bound the estimation error.

**Lemma 3** (Auer (2003), Lemma 14). *For any iteration $t \in [T]$, model $g \in \mathcal{G}$, and stage $m \in [M]$, the set of rewards $\{s_i\}_{i \in \Psi_{g,t}^m}$ are independent random variables such that $\mathbb{E}[s_i] = s_g(y_i)$.*

The second lemma shows several properties of the estimated score $\widehat{s}_{g,t}^m$ and the set $\widehat{\mathcal{G}}_t^m$. The detailed proof can be found in Appendix C.2.

**Lemma 4** (Valko et al. (2013), Lemma 7). *With probability at least $1 - (MGT)\delta$, for any iteration $t \in [T]$ and stage $m \in [M]$, the following hold:*

- $|\widehat{\mu}_{g,t}^m - s_g(y_t)| \leq (2\eta + \sqrt{\alpha})\widehat{\sigma}_{g,t}^m$ *for any $g \in \widehat{\mathcal{G}}_t^m$,*

- $\arg\max_{g \in \mathcal{G}} s_g(y_t) \in \widehat{\mathcal{G}}_t^m$, *and*

- $s_\star(y_t) - s_g(y_t) \leq 2^{3-m}$ *for any $g \in \widehat{\mathcal{G}}_t^m$.*

Now, we are ready to finish the proof of Theorem 3.

*Proof of Theorem 3.* Let $\mathcal{T}_1 := \cup_{m \in [M], g \in \mathcal{G}} \Psi_{g,T+1}^m$ and $\mathcal{T}_0 := [T] \backslash \mathcal{T}_1$. Note that $\mathcal{T}_0$ and $\mathcal{T}_1$ are sets of iterations such that model is picked in lines 8 and 13 of Algorithm 4, respectively.

**1. Regret incurred in $\mathcal{T}_0$.** For any $t \in [T]$, let $m_t$ denote the stage that model $g_t$ is picked at the $t$-th iteration. We have that

$$
\begin{aligned}
\sum_{t \in \mathcal{T}_0} (s_\star(y_t) - s_{g_t}(y_t)) &\leq \sum_{t \in \mathcal{T}_0} (\widehat{s}_{g_\star,t,t}^{m_t}(y) - s_{g_t}(y_t)) \\
&\leq \sum_{t \in \mathcal{T}_0} (\widehat{s}_{g_t,t}^{m_t}(y) - s_{g_t}(y_t)) \\
&= \sum_{t \in \mathcal{T}_0} \left(\widehat{\mu}_{g_t,t}^{m_t} + (2\eta + \sqrt{\alpha})\widehat{\sigma}_{g_t,t}^{m_t} - s_{g_t}(y_t)\right) \\
&\leq 2(2\eta + \sqrt{\alpha}) \sum_{t \in \mathcal{T}_0} \widehat{\sigma}_{g_t,t}^{m_t} \\
&\leq 2(2\eta + \sqrt{\alpha}) \sum_{t \in \mathcal{T}_0} T^{-\frac{1}{2}} \leq 2(2\eta + \sqrt{\alpha})\sqrt{T},
\end{aligned}
\tag{7}
$$

where the second inequality holds by the definition of $g_t$ and the fact that $g_{\star,t} \in \widehat{\mathcal{G}}_t^{m_t}$, and the fifth inequality holds by line 7 of Algorithm 4.

**2. Regret incurred in $\mathcal{T}_1$.**

$$
\begin{aligned}
\sum_{t \in \mathcal{T}_1} (s_\star(y_t) - s_{g_t}(y_t)) &= \sum_{g \in \mathcal{G}} \sum_{m \in [M]} \sum_{t \in \Psi_{g,T+1}^m} (s_\star(y_t) - s_{g_t}(y_t)) \\
&\leq \sum_{g \in \mathcal{G}} \sum_{m \in [M]} 2^{3-m} \cdot |\Psi_{g,T+1}^m|,
\end{aligned}
\tag{8}
$$

where the inequality holds by the last statement in Lemma 4. It remains to bound $|\Psi_{g,T+1}^m|$. First note that for any $m \in [M]$, we have that

$$
(2\eta + \sqrt{\alpha}) \sum_{t \in \Psi_{g,T+1}^m} \widehat{\sigma}_{g,t}^m > 2^{-m} \cdot |\Psi_{g,T+1}^m|
$$

from line 13 of Algorithm 4. In addition, by a similar statement of (Valko et al., 2013, Lemma 4), which is stated in Lemma 10, we have that

$$
\sum_{t \in \Psi_{g,T+1}^m} \widehat{\sigma}_{g,t}^m \leq \widetilde{O}\left(\sqrt{d_{\text{eff}}\left(10 + \frac{15}{\alpha}\right)|\Psi_{g,T+1}^m|}\right),
\tag{9}
$$

where $d_{\text{eff}}$ is defined therein and logarithmic factors are hidden in the notation $\widetilde{O}(\cdot)$. Plugging in Equation (8) results in

$$
\begin{aligned}
\sum_{t \in \mathcal{T}_1} (s_\star(y_t) - s_{g_t}(y_t)) \leq & \widetilde{O} \left( (1 + \sqrt{\alpha}) \sum_{g \in \mathcal{G}} \sum_{m \in [M]} \sqrt{d_{\text{eff}} \left( 10 + \frac{15}{\alpha} \right) |\Psi_{g,T+1}^m|} \right) \\
\leq & \widetilde{O} \left( (1 + \sqrt{\alpha}) \sqrt{GM} \sqrt{d_{\text{eff}} \left( 10 + \frac{15}{\alpha} \right) \sum_{g \in \mathcal{G}} \sum_{m \in [M]} |\Psi_{g,T+1}^m|} \right) \quad (10) \\
\leq & \widetilde{O} \left( (1 + \sqrt{\alpha}) \sqrt{d_{\text{eff}} \left( 10 + \frac{15}{\alpha} \right) GT} \right),
\end{aligned}
$$

where the second inequality holds by Cauchy-Schwarz inequality.

**3. Putting everything together.**   Combining Inequalities (7) and (10) leads to

$$
\text{Regret}(T) = \left( \sum_{t \in \mathcal{T}_0} + \sum_{t \in \mathcal{T}_1} \right) (s_\star(y_t) - s_{g_t}(y_t)) \leq \widetilde{O} \left( (1 + \sqrt{\alpha}) \sqrt{d_{\text{eff}} \left( 10 + \frac{15}{\alpha} \right) GT} \right),
$$

which concludes the proof. $\qquad \square$

# B   PROOF IN SECTION 5.2

## B.1   ERROR OF KRR ESTIMATORS WITH RANDOM FEATURES

**Theorem 4.** *Assume the error thresholds input to Algorithm 3 satisfy that $\Delta_{\text{RFF}}, \epsilon_{\text{RFF}} \leq 1/2$. Under the same conditions in Lemma 1, with probability at least $1 - 2\delta$, the quantity $\widetilde{\mu}_g$ computed by function* COMPUTE_UCB_RFF *satisfies that*

$$
|\widetilde{\mu}_g - s_g(y)| \leq \mathcal{B}_{g,1} + (2\eta + \sqrt{\alpha})(\widetilde{\sigma}_g + \mathcal{B}_{g,2}),
$$

*with the number of features satisfying Inequality (12) and bonus terms $\mathcal{B}_{g,1}$ and $\mathcal{B}_{g,2}$ given by Equations (11) and (13), where $\eta = \sqrt{2 \log(2/\delta)}$. Hence, it holds that $\widetilde{s}_g = \widetilde{\mu}_g + \mathcal{B}_{g,1} + (2\eta + \sqrt{\alpha})(\widetilde{\sigma}_g + \mathcal{B}_{g,2}) \geq s_g(y)$.*

*Proof.* The proof is based on the following two lemmas, which analyze the concentration error of the quantities $\widetilde{\mu}_g$ and $\widetilde{\sigma}_g$. The detailed proof can be found in Appendix B.3 and B.4, respectively.

**Lemma 5** (Concentration of mean using RFF)**.** *Let $\Delta_{\text{RFF}}, \epsilon_{\text{RFF}} \leq 1/2$. Under the same conditions in Lemma 1, with probability at least $1 - \delta$, the quantity $\widetilde{\mu}_g$ computed by function* COM-PUTE_UCB_RFF *satisfies that*

$$
|\widetilde{\mu}_g - \widehat{\mu}_g| \leq \mathcal{B}_{g,1} := \alpha^{-1} |\Psi_g| \, \epsilon_{\text{RFF}} + \alpha^{-2} \, \Delta_{\text{RFF}} (\|K\|_2 + \alpha) \quad (11)
$$

*with number of features*

$$
s \geq \max \left\{ \frac{4(d+2)}{\epsilon_{\text{RFF}}^2} \log \left( \frac{\sigma_p^2}{(\delta/2) \cdot \epsilon_{\text{RFF}}^2} \cdot 2^8 \right), \frac{8|\Psi_g|}{3\alpha} \Delta_{\text{RFF}}^{-2} \log \left( \frac{32 s_\alpha(K)}{\delta} \right) \right\}, \quad (12)
$$

*where $\widehat{\mu}_g = (\phi(y))^\top \Phi_g^\top (K + \alpha I)^{-1} v$, and $\sigma_p^2$ and $s_\alpha(\cdot)$ are two quantities defined in Lemmas 8 and 9, respectively.*

**Lemma 6** (Concentration of variance using RFF)**.** *Let $c > 0$ denote a lower bound of $1 - \|k_y\|_{(K+\alpha I)^{-1}}^2$. Then, conditioned on the successful events in Lemma 5, the quantity $\widetilde{\sigma}_g$ computed by function* COMPUTE_UCB_RFF *satisfies that*

$$
|\widetilde{\sigma}_g - \widehat{\sigma}_g| \leq \mathcal{B}_{g,2} := (c \cdot \alpha)^{-\frac{1}{2}} \left( 2|\Psi_g| \alpha^{-2} \, \Delta_{\text{RFF}} (\|K\|_2 + \alpha) + 3\alpha^{-1} |\Psi_g| \, \epsilon_{\text{RFF}} \right) \quad (13)
$$

*where $\widehat{\sigma}_g = \alpha^{-\frac{1}{2}} \sqrt{k(y,y) - k_y^\top (K + \alpha I)^{-1} k_y}$.*

Finally, combining Lemmas 1, 5, and 6, we derive that

$$
\begin{aligned}
|\widetilde{\mu}_g - s_g(y)| &\leq |\widetilde{\mu}_g - \widehat{\mu}_g| + |\widehat{\mu}_g - s_g(y)| \\
&\leq \mathcal{B}_{g,1} + (2\eta + \sqrt{\alpha})\widehat{\sigma}_g \\
&\leq \mathcal{B}_{g,1} + (2\eta + \sqrt{\alpha})(\widetilde{\sigma}_g + \mathcal{B}_{g,2}),
\end{aligned}
$$

which concludes the proof. $\qquad\square$

### B.2 Sup-SCK-UCB with Random Fourier Features

**Algorithm description.** To apply RFF to Sup-SCK-UCB, we replace function COMPUTE_UCB with COMPUTE_UCB_RFF in Algorithm 4. To achieve the regret bound (6), an important problem is to design (adaptive) error thresholds, i.e., $\epsilon_{\mathrm{RFF}}$ and $\Delta_{\mathrm{RFF}}$, when computing UCB at each stage $m$ and iteration $t$. We prove the regret bound in the following theorem.

**Theorem 5** (Regret of Sup-RFF-UCB). *Under Assumption 1, with probability at least $1 - \delta$, Sup-RFF-UCB attains the regret bound (6), where $\eta = \sqrt{2\log(4(\log T)GT/\delta)}$ and sequence of error thresholds input to function COMPUTE_UCB_RFF satisfying Equation (14).*

*Proof.* The proof is similar to the proof of Theorem 3. First, combining Lemmas 8 and 3, the following lemma can be proved by the exact same analysis for Lemma 4.

**Lemma 7.** *With probability at least $1 - (2MGT)\delta$, for any iteration $t \in [T]$ and stage $m \in [M]$, the following hold:*

- $|\widetilde{\mu}_{g,t}^m - s_g(y_t)| \leq \mathcal{B}_{g,1,t}^m + (2\eta + \sqrt{\alpha})(\widetilde{\sigma}_{g,t}^m + \mathcal{B}_{g,2,t}^m)$ *for any $g \in \widehat{\mathcal{G}}_t^m$,*

- $\arg\max_{g \in \mathcal{G}} s_g(y_t) \in \widehat{\mathcal{G}}_t^m$, *and*

- $s_\star(y_t) - s_g(y_t) \leq 2^{3-m}$ *for any $g \in \widehat{\mathcal{G}}_t^m$.*

*where the first statement is guaranteed by Theorem 4, $\mathcal{B}_{g,1,t}^m$ and $\mathcal{B}_{g,2,t}^m$ are the bonus (11) and (13) computed at the $m$-th stage of iteration $t$.*

Next, for iterations in $\mathcal{T}_0$ (model $g_t$ is picked in line 8 of Algorithm 4), we still have

$$
\sum_{t \in \mathcal{T}_0} (s_\star(y_t) - s_{g_t}(y_t)) \leq \widetilde{O}(\sqrt{\alpha T}).
$$

Further, for iterations in $\mathcal{T}_1$ (model $g_t$ is picked in line 13 of Algorithm 4), the third statement in the above lemma and line 14 of Algorithm 3 ensure that

$$
\begin{aligned}
&\sum_{t \in \mathcal{T}_1} (s_\star(y_t) - s_{g_t}(y_t)) \\
&\leq \sum_{g \in \mathcal{G}} \sum_{m \in [M]} 2^{3-m} \cdot |\Psi_{g,T+1}^m| \\
&< 8 \sum_{g \in \mathcal{G}} \sum_{m \in [M]} \sum_{t \in \Psi_{g,T+1}^m} \left( \mathcal{B}_{g_t,1,t}^{m_t} + (2\eta + \sqrt{\alpha}) \left( \widehat{\sigma}_{g,t}^m + \mathcal{B}_{g_t,2,t}^{m_t} \right) \right) \\
&\leq 8 \sum_{g \in \mathcal{G}} \sum_{m \in [M]} \sum_{t \in \Psi_{g,T+1}^m} \left( \mathcal{B}_{g_t,1,t}^{m_t} + (2\eta + \sqrt{\alpha}) \left( \widehat{\sigma}_{g,t}^m + 2\mathcal{B}_{g_t,2,t}^{m_t} \right) \right) \\
&= 8 \sum_{g \in \mathcal{G}} \sum_{m \in [M]} \left( \sum_{t \in \Psi_{g,T+1}^m} \left( \mathcal{B}_{g_t,1,t}^{m_t} + 2(2\eta + \sqrt{\alpha})\mathcal{B}_{g_t,2,t}^{m_t} \right) + (2\eta + \sqrt{\alpha}) \sum_{t \in \Psi_{g,T+1}^m} \widehat{\sigma}_{g,t}^m \right).
\end{aligned}
$$

Note that the upper bound of the second term has been derived in Equation (9). It remains to bound the first term. Essentially, we will find a sequence of error thresholds, and hence the number of

features defined in Inequality (12), such that the first term is bounded by $\widetilde{O}(\sqrt{GT})$. For convenience, we introduce the following notations:

**Additional notations.** For any iteration $t \in [T]$, model $g \in \mathcal{G}$, and stage $m \in [M]$, we define $K_{g,t}^m := \Phi_{g,t}^m (\Phi_{g,t}^m)^\top$, where $\Phi_{g,t}^m := [\phi(y_i)^\top]_{i \in \Psi_{g,t}^m}$. In addition, we denote by $0 < c_{g,t}^m \le 1$ the lower bound in Lemma 6 corresponding to $y_t$ and $K_{g,t}^m$. Let

$$\epsilon_{\text{RFF},g,t}^m \le t^{-\frac{1}{2}}(|\Psi_{g,t}^m|)^{-1}\sqrt{G \cdot c_{g,t}^m}, \quad \Delta_{\text{RFF},g,t}^m \le t^{-\frac{1}{2}}(|\Psi_{g,t}^m|(\|K_{g,t}^m\|_2 + \alpha))^{-1}\sqrt{G \cdot c_{g,t}^m} \quad (14)$$

denote the (upper bound of) error thresholds input to function COMPUTE_UCB_RFF.

$$\sum_{g \in \mathcal{G}, m \in [M]} \sum_{t \in \Psi_{g,T+1}^m} \mathcal{B}_{g_t,1,t}^m$$

$$= \sum_{g \in \mathcal{G}, m \in [M]} \sum_{t \in \Psi_{g,T+1}^m} \left( \alpha^{-1}|\Psi_{g,t}^m|\epsilon_{\text{RFF},g,t}^m + \alpha^{-2}\Delta_{\text{RFF},g,t}^m(\|K_{g,t}^m\|_2 + \alpha) \right)$$

$$\le \sqrt{G} \sum_{g \in \mathcal{G}, m \in [M]} \sum_{t \in \Psi_{g,T+1}^m} (\alpha^{-1}t^{-\frac{1}{2}} + \alpha^{-2}t^{-\frac{1}{2}}) \quad (15)$$

$$\le \sqrt{G} \sum_{t=1}^{T} (\alpha^{-1}t^{-\frac{1}{2}} + \alpha^{-2}t^{-\frac{1}{2}})$$

$$\le O\left( (\alpha^{-1} + \alpha^{-2})\sqrt{GT} \right)$$

where the first inequality holds by the fact that each $t \in [T]$ appears in at most one index set.

$$\sum_{g \in \mathcal{G}, m \in [M]} \sum_{t \in \Psi_{g,T+1}^m} \mathcal{B}_{g_t,2,t}^m$$

$$\le \sum_{g \in \mathcal{G}, m \in [M]} \sum_{t \in \Psi_{g,T+1}^m} (c \cdot \alpha)^{-\frac{1}{2}} \left( 2|\Psi_g|\alpha^{-2}\Delta_{\text{RFF},g,t}^m(\|K_{g,t}^m\|_2 + \alpha) + 3\alpha^{-1}|\Psi_{g,t}^m|\epsilon_{\text{RFF},g,t}^m \right)$$

$$\le \sqrt{G} \sum_{g \in \mathcal{G}, m \in [M]} \sum_{t \in \Psi_{g,T+1}^m} \left( 2\alpha^{-\frac{5}{2}}t^{-\frac{1}{2}} + 3\alpha^{-\frac{3}{2}}t^{-\frac{1}{2}} \right) \quad (16)$$

$$\le \sqrt{G} \sum_{t=1}^{T} \left( 2\alpha^{-\frac{5}{2}}t^{-\frac{1}{2}} + 3\alpha^{-\frac{3}{2}}t^{-\frac{1}{2}} \right)$$

$$\le O\left( (\alpha^{-\frac{5}{2}} + \alpha^{-\frac{3}{2}})\sqrt{GT} \right)$$

Therefore, we conclude the proof. $\qquad\square$

### B.3 Proof of Lemma 5

*Proof.* For convenience, we define $\tilde{k}_y := \widetilde{\Phi}_g(\varphi(y)) \in \mathbb{R}^{|\Psi_g|}, Q := (K + \alpha I)^{-1} \in \mathbb{R}^{|\Psi_g| \times |\Psi_g|}$, and $\widetilde{Q} := (\widetilde{K} + \alpha I)^{-1} \in \mathbb{R}^{|\Psi_g| \times |\Psi_g|}$, where $\widetilde{K} := \widetilde{\Phi}_g \widetilde{\Phi}_g^\top$. Using the same notations in the proof of Lemma 1, we obtain that

$$|\widetilde{\mu}_g - \widehat{\mu}_g| = \left| (\varphi(y))^\top \widetilde{\Phi}_g^\top (\widetilde{K} + \alpha I)^{-1}v - k_y^\top (K + \alpha I)^{-1}v \right|$$

$$= \left| \tilde{k}_y^\top \widetilde{Q}v - k_y^\top Qv \right| \quad (17)$$

$$\le \left| \tilde{k}_y^\top (\widetilde{Q} - Q)v \right| + \left| (\tilde{k}_y - k_y)^\top Qv \right|,$$

where we use Equation (22) to derive $\widetilde{\mu}_g = (\varphi(y))^\top \widetilde{\Phi}_g^\top (\widetilde{\Phi}_g^\top \widetilde{\Phi}_g + \alpha I)^{-1}\widetilde{\Phi}_g^\top v = (\varphi(y))^\top \widetilde{\Phi}_g^\top (\widetilde{K} + \alpha I)^{-1}v$ in the first equation.

**1. Bounding $|(\tilde{k}_y - k_y)^\top Qv|$.** We evoke (Rahimi & Recht, 2007a, Claim 1), which is rewritten in Lemma 8 using our notations. For a desired threshold $\epsilon_{\text{RFF}} > 0$, set

$$s = \frac{4(d+2)}{\epsilon_{\text{RFF}}{}^2} \log \left( \frac{\sigma_p^2}{(\delta/2) \cdot \epsilon_{\text{RFF}}{}^2} \cdot 2^8 \right).$$

Then, with probability at least $1 - \frac{\delta}{2}$, it holds that $\sup_{y,y' \in \mathcal{Y}} |(\varphi(y))^\top \varphi(y') - k(y, y')| \leq \epsilon_{\text{RFF}}$, and hence $\|\tilde{k}_y - k_y\|_\infty \leq \epsilon_{\text{RFF}}$. Therefore, we obtain

$$|(\tilde{k}_y - k_y)^\top Qv| \leq \|\tilde{k}_y - k_y\|_2 \cdot \|Q\|_2 \cdot \|v\|_2 \leq \epsilon_{\text{RFF}} \sqrt{|\Psi_g|} \cdot \alpha^{-1} \cdot \sqrt{|\Psi_g|} = \alpha^{-1} |\Psi_g| \epsilon_{\text{RFF}}, \quad (18)$$

where the last inequality holds by $\|Q\|_2 = \lambda_{\min}^{-1}(K + \alpha I) \leq \alpha^{-1}$ and $\|v\|_\infty \leq 1$.

**2. Bounding $|\tilde{k}_y^\top (\widetilde{Q} - Q)v|$.** Note that

$$
\begin{aligned}
|\tilde{k}_y^\top (\widetilde{Q} - Q)v| &\leq \|\tilde{k}_y\|_2 \cdot \|\widetilde{Q} - Q\|_2 \cdot \|v\|_2 \\
&\leq \sqrt{2|\Psi_g|} \cdot \|\widetilde{Q} - Q\|_2 \cdot \sqrt{|\Psi_g|}
\end{aligned}
$$

where the first inequality holds by the fact that $(\varphi(y))^\top \varphi(y_i) = (2/s) \sum_{j=1}^s \cos(w_j^\top y + b_j) \cos(w_{i,j}^\top + b_{i,j}) \leq 2$. To bound $\|\widetilde{Q} - Q\|_2$, we evoke (Avron et al., 2017, Theorem 7), which is rewritten in Lemma 9. For a desired threshold $\Delta_{\text{RFF}} \leq 1/2$, the following inequality holds with probability at least $1 - \frac{\delta}{2}$:

$$(1 - \Delta_{\text{RFF}})(K + \alpha I) \preceq \widetilde{K} + \alpha I$$

for $s \geq \frac{8|\Psi_g|}{3\alpha} \Delta_{\text{RFF}}{}^{-2} \log(32 s_\alpha(K)/\delta)$. By Sherman-Morrison-Woodbury formula, i.e., $A^{-1} - B^{-1} = A^{-1}(B - A)B^{-1}$ where $A$ and $B$ are invertible, we derive

$$
\begin{aligned}
&\|(\widetilde{K} + \alpha I)^{-1} - (K + \alpha I)^{-1}\|_2 \\
&\leq \|(\widetilde{K} + \alpha I)^{-1}\|_2 \cdot \|(K + \alpha I) - (\widetilde{K} + \alpha I)\|_2 \cdot \|(K + \alpha I)^{-1}\|_2 \\
&\leq \alpha^{-2} \Delta_{\text{RFF}}(\|K\|_2 + \alpha)
\end{aligned}
\quad (19)
$$

where the last inequality holds by the fact that $\|(\widetilde{K} + \alpha I)^{-1}\|_2, \|(K + \alpha I)^{-1}\|_2 \leq \alpha^{-1}$ and $\|(K + \alpha I) - (\widetilde{K} + \alpha I)\|_2 \leq \|\Delta_{\text{RFF}}(K + \alpha I)\|_2 \leq \Delta_{\text{RFF}}(\|K\|_2 + \alpha)$.

**3. Putting everything together.** Combining Equations (18) and (19), with probability at least $1 - \delta$, it holds that

$$|\widetilde{\mu}_g - \widehat{\mu}_g| \leq \alpha^{-1}|\Psi_g| \epsilon_{\text{RFF}} + \alpha^{-2} \Delta_{\text{RFF}}(\|K\|_2 + \alpha)$$

when

$$s \geq \max \left\{ \frac{4(d+2)}{\epsilon_{\text{RFF}}{}^2} \log \left( \frac{\sigma_p^2}{(\delta/2) \cdot \epsilon_{\text{RFF}}{}^2} \cdot 2^8 \right), \frac{8|\Psi_g|}{3\alpha} \Delta_{\text{RFF}}{}^{-2} \log \left( \frac{32 s_\alpha(K)}{\delta} \right) \right\},$$

which concludes the proof. $\qquad\square$

### B.4 PROOF OF LEMMA 6

*Proof.* We use the same notations in the proof of Lemma 5. Let $c$ denote a lower bound of $1 - \|k_y\|^2_{(K+\alpha I)^{-1}}$. Note that

$$
\begin{aligned}
&|\widetilde{\sigma}_g - \widehat{\sigma}_g| \\
&= \alpha^{-\frac{1}{2}} \left| \sqrt{1 - (\varphi(y))^\top \widetilde{\Phi}_g^\top (\widetilde{K} + \alpha I)^{-1} \widetilde{\Phi}_g(\varphi(y))} - \sqrt{1 - k_y^\top (K + \alpha I)^{-1} k_y} \right| \\
&\le (c \cdot \alpha)^{-\frac{1}{2}} \left| (\varphi(y))^\top \widetilde{\Phi}_g^\top (\widetilde{K} + \alpha I)^{-1} \widetilde{\Phi}_g(\varphi(y)) - k_y^\top (K + \alpha I)^{-1} k_y \right| \\
&= (c \cdot \alpha)^{-\frac{1}{2}} \left| \tilde{k}_y^\top \widetilde{Q} \tilde{k}_y - k_y^\top Q k_y \right| \\
&\le (c \cdot \alpha)^{-\frac{1}{2}} \left( \left| \tilde{k}_y^\top (\widetilde{Q} - Q) \tilde{k}_y \right| + \left| (\tilde{k}_y - k_y)^\top Q \tilde{k}_y \right| + \left| k_y^\top Q (\tilde{k}_y - k_y) \right| \right) \\
&\le (c \cdot \alpha)^{-\frac{1}{2}} \left( \|\tilde{k}_y\|_2^2 \|\widetilde{Q} - Q\|_2 + \|\tilde{k}_y - k_y\|_2 \|\tilde{k}_y\|_2 \|Q\|_2 + \|\tilde{k}_y - k_y\|_2 \|k_y\|_2 \|Q\|_2 \right) \\
&\le (c \cdot \alpha)^{-\frac{1}{2}} \left( 2|\Psi_g| \alpha^{-2} \Delta_{\text{RFF}} (\|K\|_2 + \alpha) + 3\alpha^{-1} |\Psi_g| \epsilon_{\text{RFF}} \right)
\end{aligned}
\tag{20}
$$

which concludes the proof. $\square$

### B.5 ANALYSIS OF LEMMA 2

*Proof.* Solving KRR with $n$ regression data requires $\Theta(n^3)$ time and $\Theta(n^2)$ space. Hence, by the convexity of the cubic and quadratic functions, the time for COMPUTE_UCB scales with $\Theta(\sum_{g \in \mathcal{G}} n_g^3) = O(t^3/G^2)$, and the space scales with $\Theta(\sum_{g \in \mathcal{G}} n_g^2) = O(t^2/G)$, where $n_g := |\Psi_g|$ is the visitation to any model $g \in \mathcal{G}$ up to iteration $t$, and we have $\sum_{g \in \mathcal{G}} n_g = t$. On the other hand, solving KRR with $n$ regression data and random features of size $s$ requires $O(ns^2)$ time and $O(ns)$ space. Therefore, the time for COMPUTE_UCB_RFF scales with $O(\sum_{g \in \mathcal{G}} n_g s^2) = O(ts^2)$, and the space scales with $O(\sum_{g \in \mathcal{G}} n_g s) = O(ts)$. $\square$

## C AUXILIARY LEMMAS

### C.1 PROOF OF LEMMA 1

*Proof.* We rewrite the proof using the notations in Section 5. Obviously, Equation (5) holds when the index set $\Psi_g$ is empty. In the following, we consider non-empty $\Psi_g$. Let $\Phi_g := [\phi(y_i)^\top]_{i \in \Psi_g}$. Note that $k_y = [k(y, y_i)]_{i \in \Psi_g}^\top = \Phi_g(\phi(y))$ and $K = [k(y_i, y_j)]_{i,j \in \Psi_g} = \Phi_g \Phi_g^\top$. We have

$$
\begin{aligned}
\widehat{\mu}_g - s_g(y) &= (\phi(y))^\top \Phi_g^\top (K + \alpha I)^{-1} v - (\phi(y))^\top w_g^\star \\
&= (\phi(y))^\top (\Phi_g^\top \Phi_g + \alpha I)^{-1} \Phi_g^\top v - (\phi(y))^\top (\Phi_g^\top \Phi_g + \alpha I)^{-1} (\Phi_g^\top \Phi_g + \alpha I) w_g^\star \quad (21) \\
&= (\phi(y))^\top (\Phi_g^\top \Phi_g + \alpha I)^{-1} \Phi_g^\top (v - \Phi_g w_g^\star) - \alpha (\phi(y))^\top (\Phi_g^\top \Phi_g + \alpha I)^{-1} w_g^\star,
\end{aligned}
$$

where the second equation holds by the positive definiteness of both matrices $(K + \alpha I)$ and $(\Phi_g^\top \Phi_g + \alpha I)$ and hence

$$
\Phi_g^\top (K + \alpha I)^{-1} = (\Phi_g^\top \Phi_g + \alpha I)^{-1} \Phi_g^\top.
\tag{22}
$$

**1. Bounding** $(\phi(y))^\top (\Phi_g^\top \Phi_g + \alpha I)^{-1} \Phi_g^\top (v - \Phi_g w_g^\star)$. Note that the scores $\{s_t : t \in \Psi_g\}$ are independent by the construction of $\Phi_g$ and $\mathbb{E}[s_t] = (w_g^\star)^\top \phi(y_t)$, we have that

$$
(\phi(y))^\top (\Phi_g^\top \Phi_g + \alpha I)^{-1} \Phi_g^\top (v - \Phi_g w_g^\star) = \sum_{i=1}^{|\Psi_g|} [(\phi(y))^\top (\Phi_g^\top \Phi_g + \alpha I)^{-1} \Phi_g^\top]_i \cdot [v - \Phi_g w_g^\star]_i
$$

are summation of zero mean independent random variables, where we denote by $[\cdot]_i$ the $i$-th element of a vector. Further, each variable satisfies that

$$
\begin{aligned}
&\left|[(\phi(y))^\top(\Phi_g^\top\Phi_g+\alpha I)^{-1}\Phi_g^\top]_i \cdot [v-\Phi_g w_g^\star]_i\right| \\
&\leq \|(\phi(y))^\top(\Phi_g^\top\Phi_g+\alpha I)^{-1}\Phi_g^\top\| \cdot |[v-\Phi_g w_g^\star]_i| \\
&\leq \sqrt{(\phi(y))^\top(\Phi_g^\top\Phi_g+\alpha I)^{-1}\Phi_g^\top\Phi_g(\Phi_g^\top\Phi_g+\alpha I)^{-1}(\phi(y))} \cdot (1+\|w_g^\star\|) \\
&\leq 2\widehat\sigma_g
\end{aligned}
$$

where the last inequality holds by $\|w_g^\star\|\leq 1$ and the second inequality holds by

$$
\begin{aligned}
\widehat\sigma_g &= \alpha^{-\frac{1}{2}}\sqrt{k(y,y)-k_y^\top(K+\alpha I)^{-1}k_y} \\
&= \alpha^{-\frac{1}{2}}\sqrt{(\phi(y))^\top(\phi(y))-(\phi(y))^\top\Phi_g^\top(K+\alpha I)^{-1}\Phi_g(\phi(y))} \\
&= \alpha^{-\frac{1}{2}}\sqrt{(\phi(y))^\top\left(I-(\Phi_g^\top\Phi_g+\alpha I)^{-1}\Phi_g^\top\Phi_g\right)(\phi(y))} \\
&= \sqrt{(\phi(y))^\top(\Phi_g^\top\Phi_g+\alpha I)^{-1}(\phi(y))},
\end{aligned}
$$

Then, by Azuma-Hoeffding inequality, it holds that

$$
\begin{aligned}
&\mathbb{P}\left(|(\phi(y))^\top(\Phi_g^\top\Phi_g+\alpha I)^{-1}\Phi_g^\top(v-\Phi_g w_g^\star)|>2\eta\widehat\sigma_g\right) \\
&\leq 2\exp\left(-\frac{\widehat\sigma_g^2\eta^2}{2|\Psi_g|\widehat\sigma_g^2}\right) \\
&\leq 2\exp(-\eta^2/2)
\end{aligned}
\tag{23}
$$

**2. Bounding $\alpha(\phi(y))^\top(\Phi_g^\top\Phi_g+\alpha I)^{-1}w_g^\star$.**  By the Cauchy-Schwarz inequality, it holds that

$$
\begin{aligned}
&\left|(\phi(y))^\top(\Phi_g^\top\Phi_g+\alpha I)^{-1}w_g^\star\right| \\
&\leq \|w_g^\star\|\cdot\|(\phi(y))^\top(\Phi_g^\top\Phi_g+\alpha I)^{-1}\| \\
&= \|w_g^\star\|\cdot\sqrt{(\phi(y))^\top(\Phi_g^\top\Phi_g+\alpha I)^{-1}\alpha^{-1}\alpha I(\Phi_g^\top\Phi_g+\alpha I)^{-1}(\phi(y))} \\
&\leq \alpha^{-1/2}\sqrt{(\phi(y))^\top(\Phi_g^\top\Phi_g+\alpha I)^{-1}(\Phi_g^\top\Phi_g+\alpha I)(\Phi_g^\top\Phi_g+\alpha I)^{-1}(\phi(y))} \\
&= \alpha^{-1/2}\widehat\sigma_g,
\end{aligned}
\tag{24}
$$

where the second inequality holds by the positive definiteness of $\Phi_g^\top\Phi_g$.

**3. Putting everything together.**  Plugging (23) and (24) in (21) and setting $\delta=2\exp(-\eta^2/2)$ concludes the proof.  $\square$

### C.2  Proof of Lemma 4

*Proof.* The first statement holds by both Lemma 3 and Lemma 1, and a uniform bound over all $t\in[T], g\in\mathcal{G}$, and $m\in[M]$. Let $g_{\star,t}:=\arg\max_{g\in\mathcal{G}}s_g(y_t)$ is the optimal model for prompt $y_t$ and $\widehat g_{\star,t}^m:=\arg\max_{g\in\widehat{\mathcal{G}}_t^m}\widehat s_{g,t}^m$ is optimistic model at stage $m$.

To show the second statement, first note that $g_{\star,t}\in\widehat{\mathcal{G}}_t^1$. Assume $g_{\star,t}\in\widehat{\mathcal{G}}_t^m$ for some $m\in[M-1]$. Then, by the first statement, we obtain that $\widehat s_{g_{\star,t},t}^m - \max_{g\in\widehat{\mathcal{G}}_t^m}\widehat s_{g,t}^m \geq s_{g_{\star,t}}(y_t)-(2\eta+\sqrt\alpha)\cdot\widehat\sigma_{g_{\star,t}}^m - (s_{\widehat g_{\star,t}^m}(y_t)+(2\eta+\sqrt\alpha)\cdot\widehat\sigma_{\widehat g_{\star,t}^m,t}^m) \geq -(2\eta+\sqrt\alpha)\cdot(\widehat\sigma_{g_{\star,t}}^m+\widehat\sigma_{\widehat g_{\star,t}^m,t}^m) \geq 2\cdot 2^{-m}=2^{1-m}$, which ensures $g_{\star,t}\in\widehat{\mathcal{G}}_t^{m+1}$.

Finally, by the first two statements, we have that $s_\star(y_t)-s_g(y_t)\leq \widehat s_{g_{\star,t},t}^m+2(2\eta+\sqrt\alpha)\cdot\widehat\sigma_{g_{\star,t}}^m - (\widehat s_{g,t}^m-2(2\eta+\sqrt\alpha)\cdot\widehat\sigma_g^{m,t})\leq 2\cdot 2^{1-m}=2^{3-m}$. We conclude the proof.  $\square$

### C.3 USEFUL LEMMAS

**Lemma 8** (Rahimi & Recht (2007a), Claim 1). *Let $\mathcal{M}$ be a compact subset of $\mathbb{R}^d$ with diameter* diam$(\mathcal{M})$. *Then, for the mapping $\varphi : \mathbb{R}^d \to \mathbb{R}^s$ defined in Equation (3), we have*

$$\mathbb{P}\left(\sup_{y,y' \in \mathcal{M}} |(\varphi(y))^\top \varphi(y') - k(y,y')| \geq \epsilon\right) \leq 2^8 \left(\frac{\sigma_p \cdot \text{diam}(\mathcal{M})}{\epsilon}\right)^2 \exp\left(-\frac{s\epsilon^2}{4(d+2)}\right),$$

*where $\sigma_p^2 := \mathbb{E}_p[\omega^\top \omega]$ is the second moment of the Fourier transform of $k$.[7] Further,* $\sup_{y,y' \in \mathcal{M}} |(\varphi(y))^\top \varphi(y') - k(y,y')| \leq \epsilon$ *with any constant probability when $s = \Omega((d/\epsilon^2) \log(\sigma_p \cdot \text{diam}(\mathcal{M})/\epsilon))$.*

**Lemma 9** (Avron et al. (2017), Theorem 7). *Let $K = [k(y_i, y_j)]_{i,j \in [n]}$ denote the Gram matrix of $\{y_i \in \mathbb{R}^d\}_{i=1}^n$, where $k$ is a shift-invariant kernel function. Let $\Delta \leq 1/2$ and $\delta \in (0,1)$. Assume that $\|K\|_2 \geq \alpha$. If we use $s \geq \frac{8n}{3\alpha}\Delta^{-2}\log(16s_\alpha(K)/\delta)$ random Fourier features, then with probability at least $1 - \delta$, it holds that*

$$(1 - \Delta)(K + \alpha I) \preceq \widetilde{K} + \alpha I \preceq (1 + \Delta)(K + \alpha I)$$

*where $s_\alpha(K) := \text{Tr}[(K + \alpha I)^{-1} K]$ and we denote by $\widetilde{K} = [(\varphi(y_i))^\top (\varphi(y_j))]_{i,j \in [n]}$ the approximated Gram matrix using $s \in \mathbb{N}_+$ random Fourier features, where $\varphi : \mathbb{R}^d \to \mathbb{R}^s$ is the feature mapping.*

**Lemma 10** (Valko et al. (2013), Lemma 4). *For any model $g \in \mathcal{G}$ and stage $m \in [M]$, let $\lambda_{g,1}^m \geq \lambda_{g,2}^m \geq \cdots$ denote the eigenvalues (in the decreasing order) of the matrix $(\Phi_g^m)^\top \Phi_g^m + \alpha I$, where $\Phi_g^m = [\phi(y_i)^\top]_{i \in \Psi_{g,T+1}^m}$. Then, for any iteration $t \in [T]$, it holds that*

$$\sum_{t \in \Psi_{g,T+1}^m} \widehat{\sigma}_{g,t}^m \leq \widetilde{O}\left(\sqrt{d_{\text{eff}}\left(10 + \frac{15}{\alpha}\right)|\Psi_{g,T+1}^m|}\right)$$

*where $d_{\text{eff}} := \max_{g \in \mathcal{G}, m \in [M]} \min\{j \in \mathbb{N}_+ : j\alpha \log T \geq \Lambda_{g,j}^m\}$ and $\Lambda_{g,j}^m := \sum_{i>j} \lambda_{g,i}^m - \alpha$ is the effective dimension.*

## D ADDITIONAL EXPERIMENTAL DETAILS AND RESULTS

**1. Implementation details.** We use the CLIP-based features of the prompts as the context vector (Cherti et al., 2023) for tasks of T2I and T2V generation, and we use the CLIP-based features of the images in the task of image captioning. We set both the exploration and regularization parameters $\alpha, \eta = 1$ in all the experiments. Two hyperparameters have to be chosen. The first one is the parameter $\gamma$ in the polynomial and radial basis function (RBF) kernels, which are given by

$$k_{\text{poly3}}^\gamma(x_1, x_2) = (\gamma \cdot x_1^\top x_2 + 1)^3, \quad k_{\text{RBF}}^\gamma(x_1, x_2) = \exp(-\gamma \cdot \|x_1 - x_2\|^2).$$

In the experiments, we select $\gamma$ to be 5 and 1 for the polynomial and RBF kernel functions, respectively. The second hyperparameter is the number of random features in the RFF-UCB algorithm. In addition, the features are generated once for each sample and stored to save the computation of the RFF-UCB algorithm.

**2. Ablation study on hyperparameters.** We conduct ablation studies on the selections of parameter $\gamma$ in the RBF kernel function and the number of features in RFF-UCB. The results are summarized in Figures 8 and 9, respectively. We select $\gamma = 1$ (default), $3, 5$, and 7 and number of features varying between 25, 50 (default), 75, and 100. Results show that the RFF-UCB algorithm can attain consistent performance. [Additionally, we test the SCK-UCB-poly3 algorithm with $\gamma = 1, 3, 5$ (default), and 7 in the polynomial kernel and regularization parameter $\alpha = 0.5, 1.0$ (default), and 1.5 in KRR. The results are summarized in Figures 10 and 11, respectively.]

**3. Comparison on running time.** [We compare the running time of SCK-UCB and RFF-UCB, and the results are summarized in Figure 12.]

---

[7]For the RBF kernel with parameter $\sigma^2$, i.e., $k_{\text{RBF}}(y, y') = \exp(-\frac{1}{2\sigma^2}\|y - y'\|_2^2)$, we have $\sigma_{p_{\text{RBF}}}^2 = \frac{d}{\sigma^2}$.

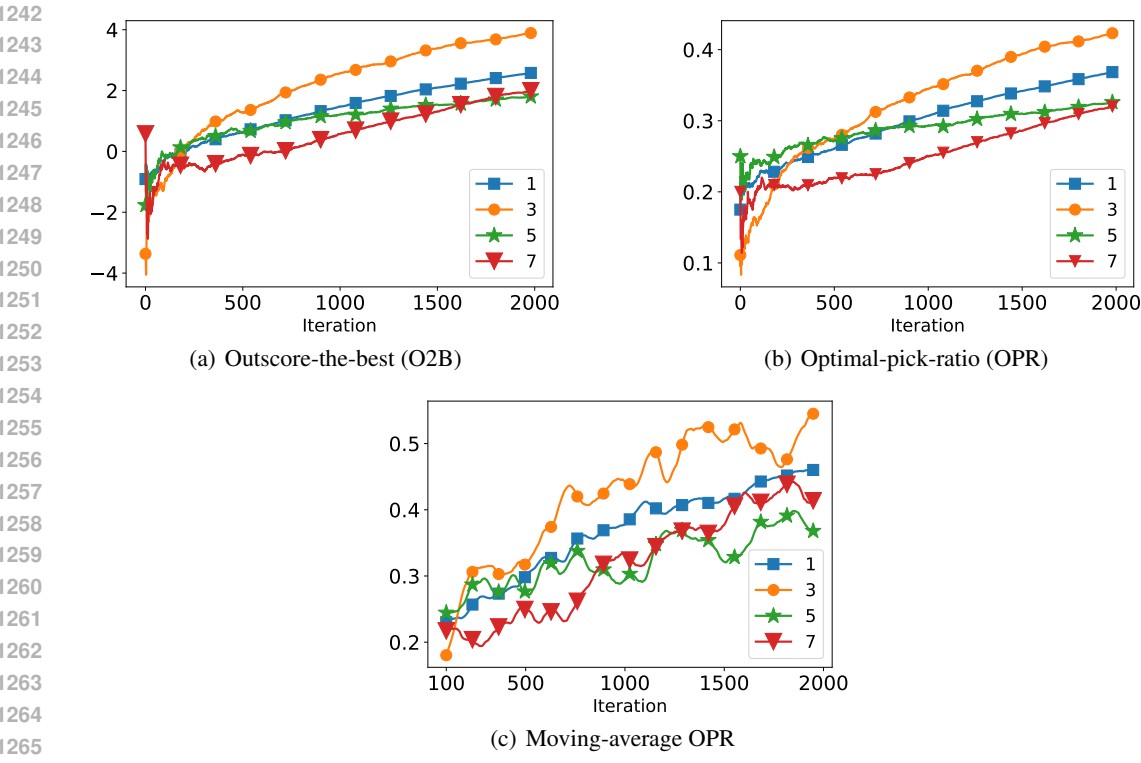

(a) Outscore-the-best (O2B)

(b) Optimal-pick-ratio (OPR)

(c) Moving-average OPR

Figure 8: Parameter $\gamma$ in the RBF kernel function: Results are averaged over 20 trials.

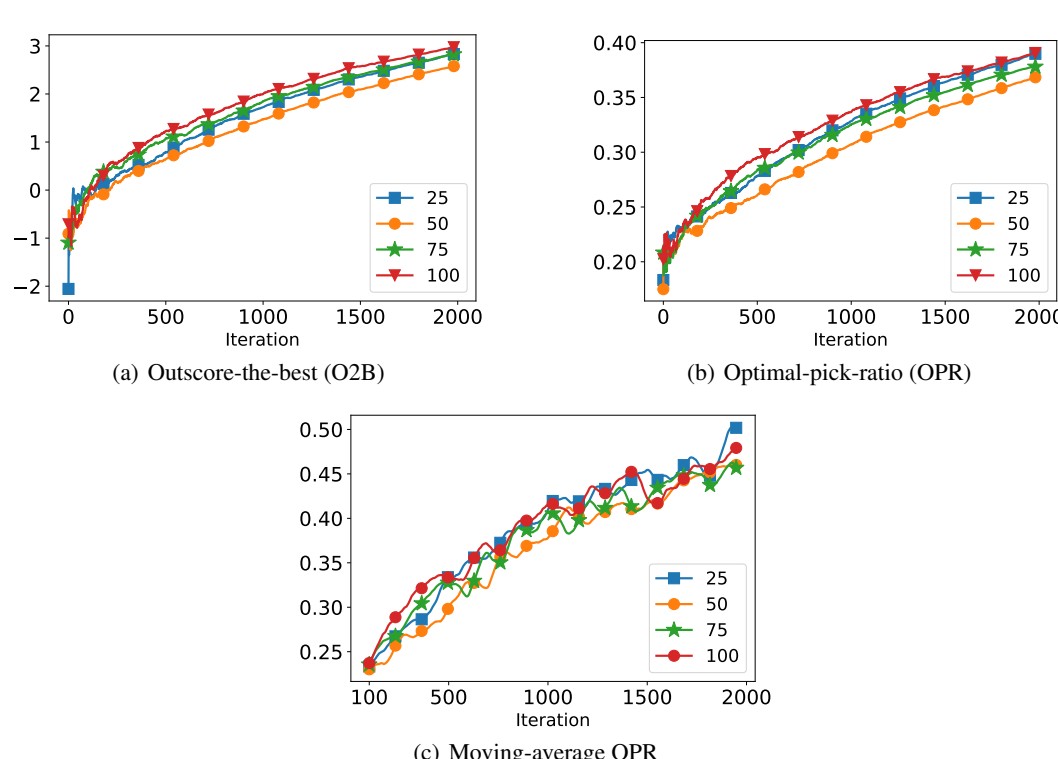

(a) Outscore-the-best (O2B)

(b) Optimal-pick-ratio (OPR)

(c) Moving-average OPR

Figure 9: Number of random features in RFF-UCB: Results are averaged over 20 trials.

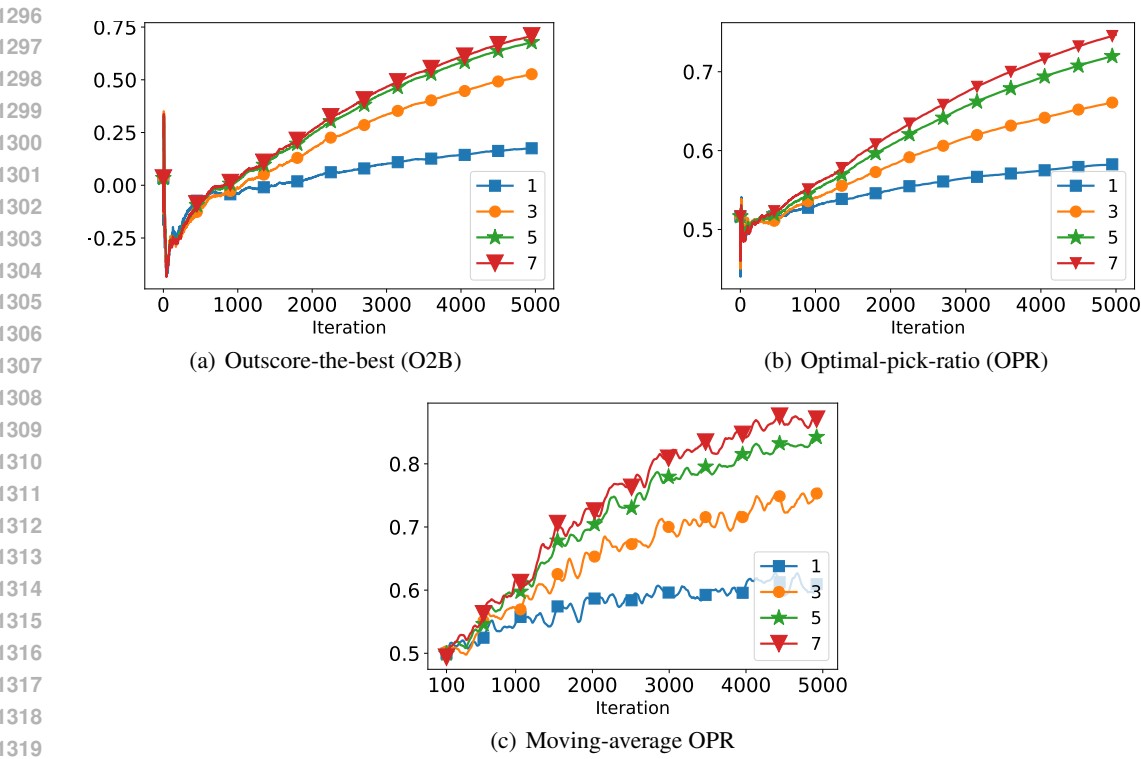

(a) Outscore-the-best (O2B)

(b) Optimal-pick-ratio (OPR)

(c) Moving-average OPR

Figure 10: Parameter $\gamma$ in the polynomial kernel function: Results are averaged over 20 trials.

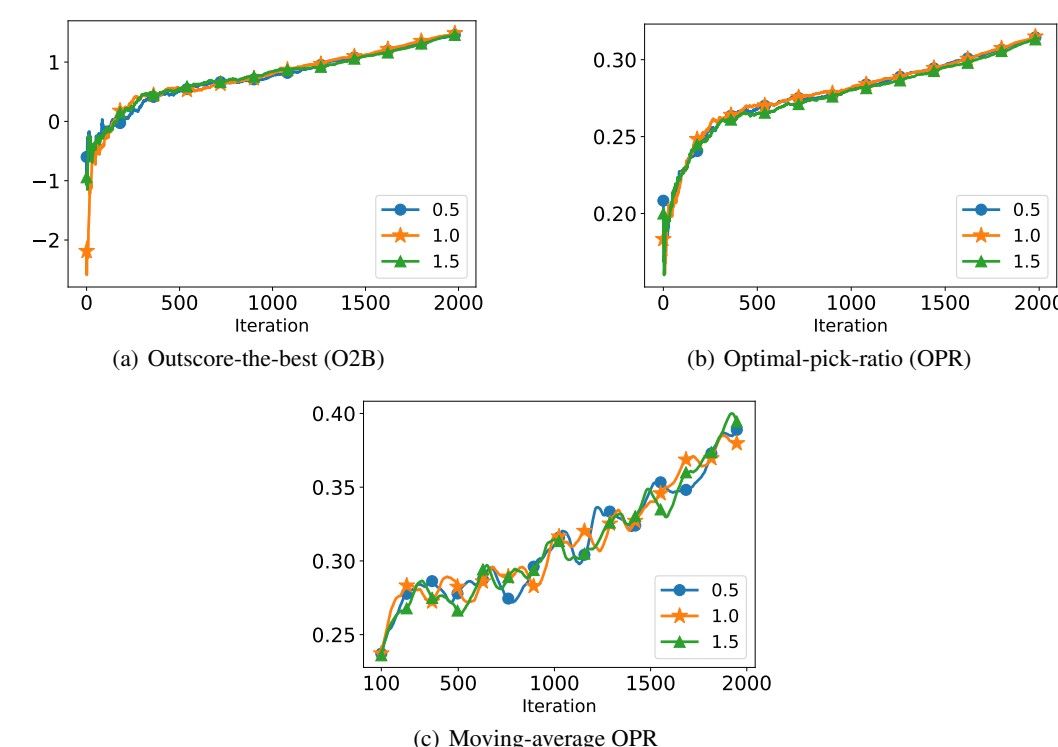

(a) Outscore-the-best (O2B)

(b) Optimal-pick-ratio (OPR)

(c) Moving-average OPR

Figure 11: Regularization parameter $\alpha$ in KRR: Results are averaged over 20 trials.

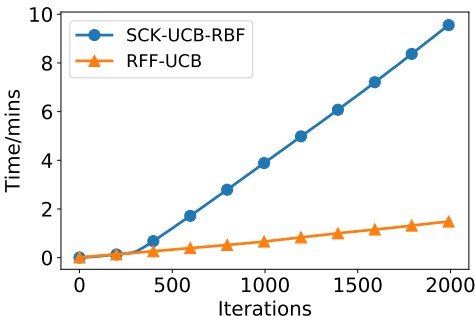

Figure 12: Running time: The execution time of SCK-UCB-RBF (SCK-UCB using the RBF kernel) and RFF-UCB on Setup 4. SCK-UCB-RBF takes around 10 minutes to finish 2,000 iterations of model selection, while RFF-UCB uses less than 2 minutes. Results are averaged over 20 trials.

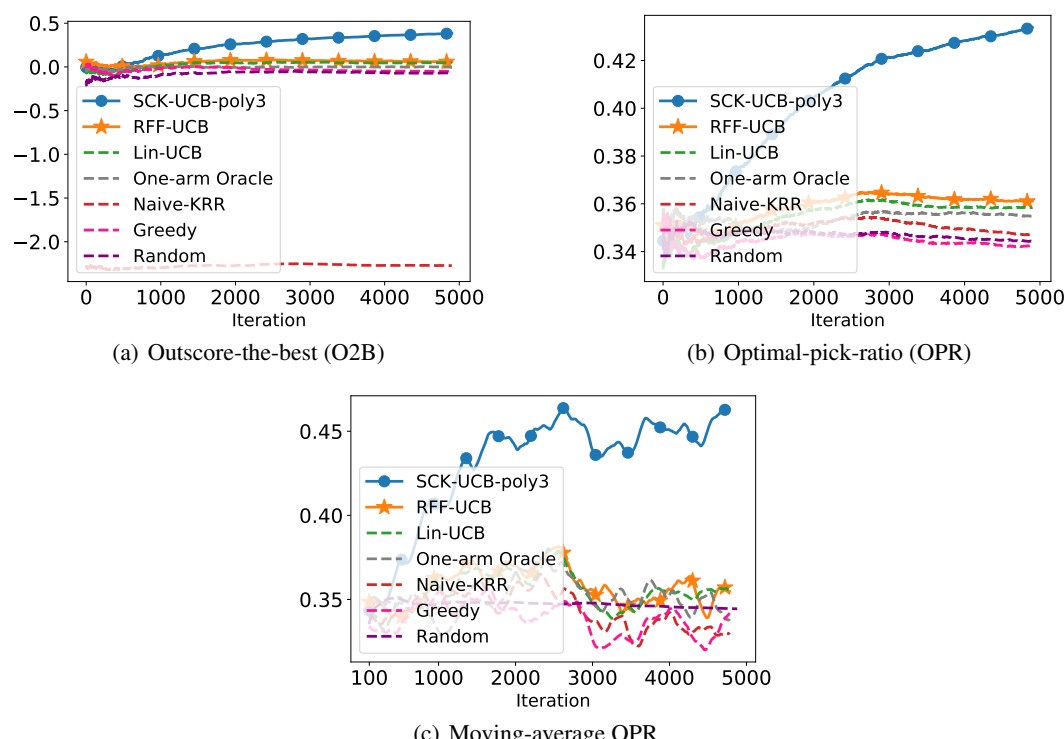

(a) Outscore-the-best (O2B)  (b) Optimal-pick-ratio (OPR)

(c) Moving-average OPR

Figure 13: T2I generation with newly-introduced prompt types: Prompts are drawn from two categories in the MS-COCO dataset for the first 1k iterations. After that, an additional prompt category is added after each 1k iterations. Images are generated from PixArt-$\alpha$ and uniDiffuser. Results are averaged over 20 trials.

**4. Additional examples.** We provide more examples showing that prompt-based generative models can outperform for text prompts from certain categories while underperforming for other text categories (see Figures 17 and 18).

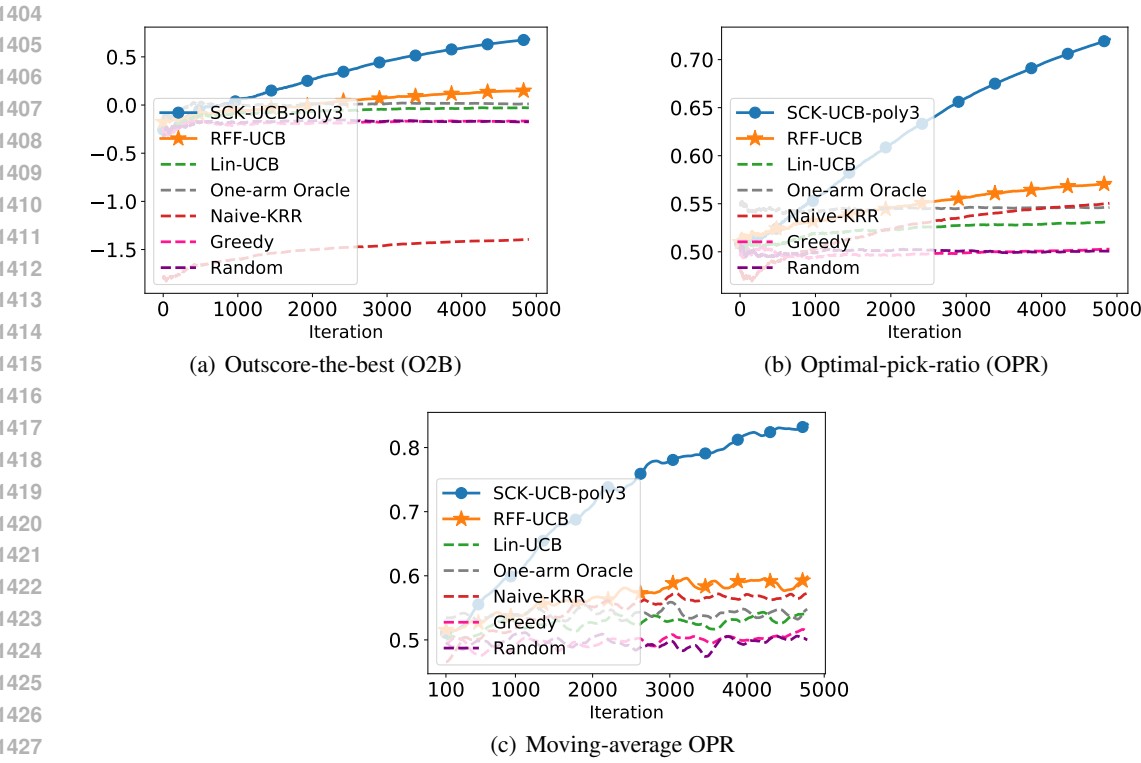

Figure 14: Prompt-based selection between uniDiffuser and PixArt-$\alpha$ (see Figure 17): Results are averaged over 20 trials.

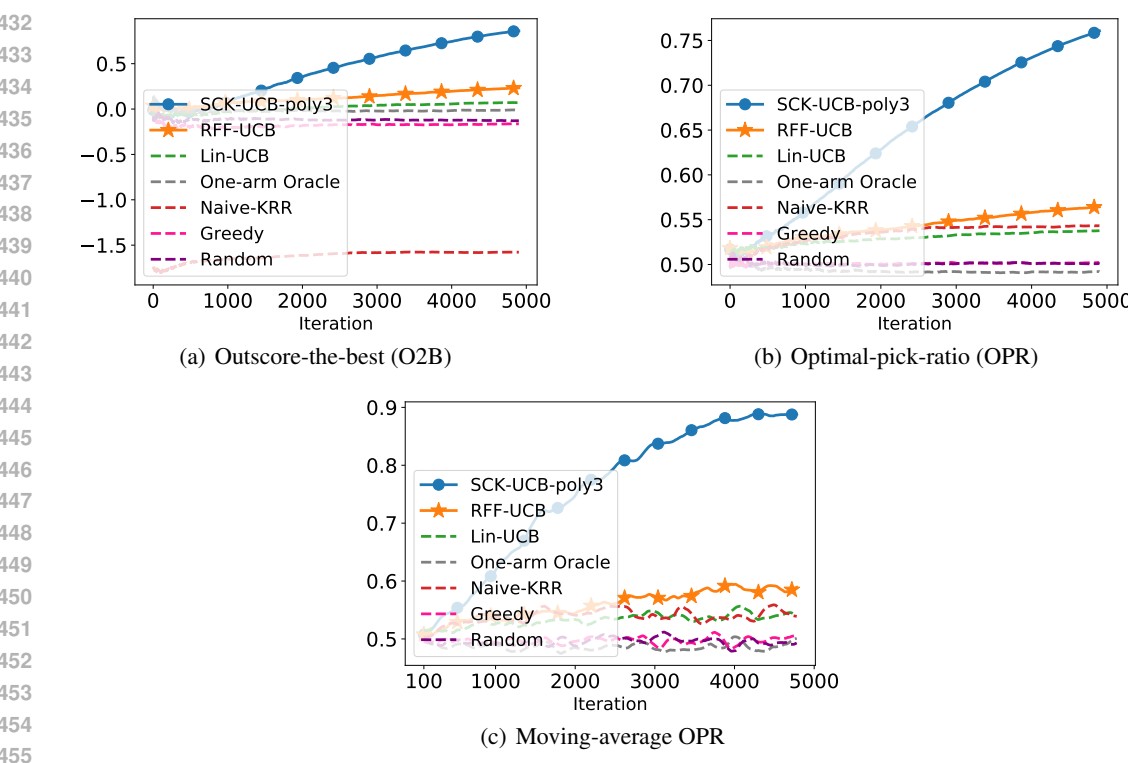

Figure 15: Prompt-based selection between uniDiffuser and Stable Diffusion (see Figure 18): Results are averaged over 20 trials.

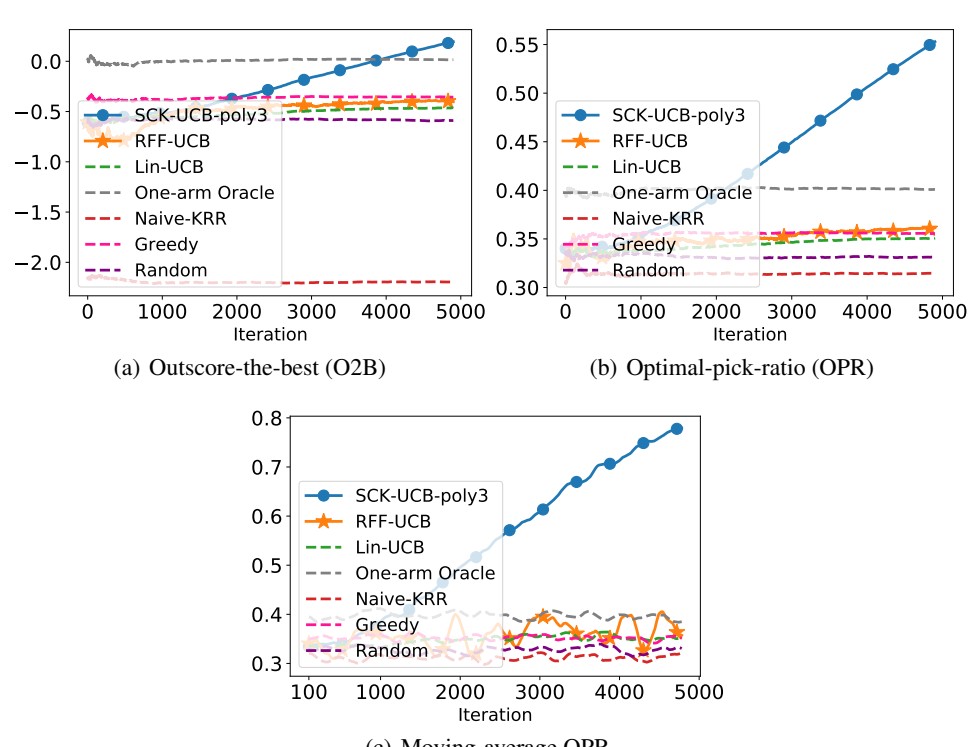

Figure 16: Prompt-based selection among Stable Diffusion v1-5, PixArt-$\alpha$, and DeepFloyd: Prompts are drawn from types "carrot" and "bowl" in the MS-COCO dataset. Results are averaged over 20 trials.

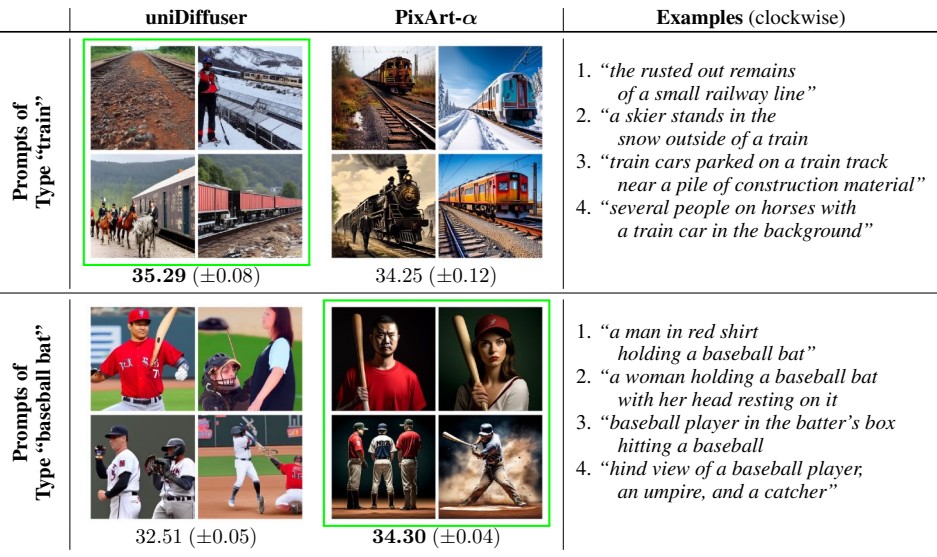

Figure 17: Prompt-based generated images from uniDiffuser (Bao et al., 2023) and PixArt-$\alpha$: uniDiffuser attains a higher CLIPScore in generating type "train" prompts (35.29 versus 34.25) while underperforms for type "baseball bat" prompts (32.51 versus 34.30).

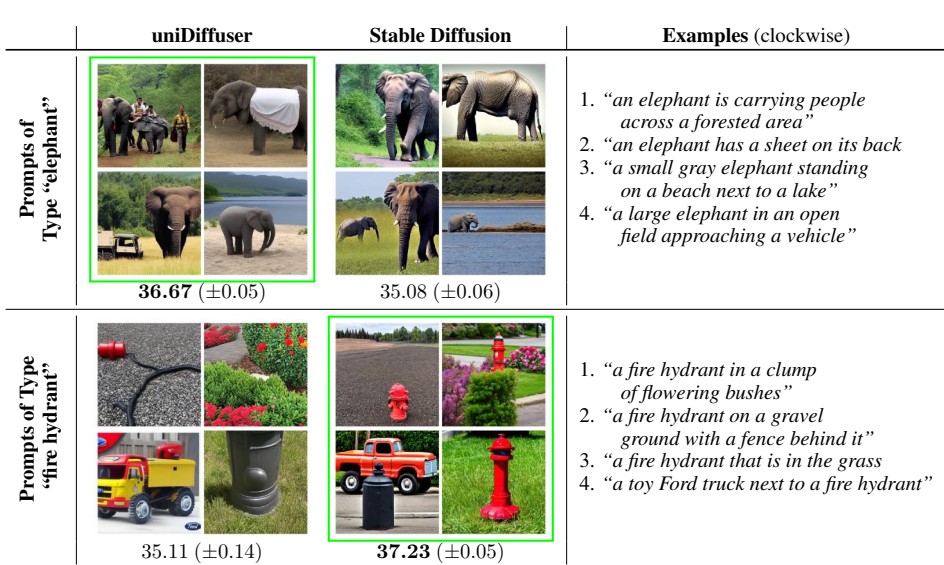

Figure 18: Prompt-based generated images from uniDiffuser and Stable Diffusion: uniDiffuser attains a higher CLIPScore in generating type "elephant" prompts (36.67 versus 35.08) while underperforms for type "fire hydrant" prompts (35.11 versus 37.23).

| | Example 1 | | Example 2 | | Example 3 | |
|---|---|---|---|---|---|---|
| **Clean** | | *"a person on a surfboard in the air above the water"* **28.57** | | *"a man standing in a kitchen with a dog"* **40.53** | | *"a bowl filled with ice cream and strawberries* **27.98** |
| **Noisy** | | *"a blurry photo of a skateboarder flying through the air"* 24.72 | | *"a cat that is standing in the grass"* 13.27 | | *"a blue and white bowl filled with water"* 25.83 |

Figure 19: Generated captions for the clean and noise-perturbed images from vit-gpt2 and the corresponding CLIPScore.

