# OpenReview forum: "An Online Learning Approach to Prompt-based Selection of Generative Models"
_ICLR.cc/2025/Conference — Submitted to ICLR 2025_

### Official Review · Reviewer_WovK · 2024-10-22

**Soundness:** 2
**Presentation:** 1
**Contribution:** 2
**Rating:** 5
**Confidence:** 3

**Summary:**

This paper studies the contextual bandit problem of selecting the best image generation models given input prompts. The work is motivated by the fact that existing work aims to identify the model that maximizes the average evaluation score across data, while picking different models for each input text may improve the evaluation metrics as different image models can work well on different prompts. For methodology, the paper adopts kernelized contextual bandits (CB) with random Fourier features (RFF) from the existing literature on contextual bandits.

**Strengths:**

- **The problem of selecting models per prompt is well-motivated.**

The idea of selecting different models for each input text rather than using a single model for the entire data is explained well. The illustrative example in Figure 1, which shows that the superiority of two models can change with two types of images, also nicely highlights the existing issues.

- **Experiments on real data.**

The experiments are conducted using two image generation models. It is good that the performance is verified on real data, not only the synthetic data with noises.

**Weaknesses:**

- **Is linear bandits really infeasible in this setting (i.e., is kernel CB is needed)?**

The paper argues that the paper proposes kernel CB because "the relationship between the prompt vector and score is often highly non-linear and generator-dependent". However, when using the clip score, I could not agree with this statement because the clip score can be (nearly) explained with the linear model as $s = \max ( 0, 100 \cdot cos(v_x, c_y) ) = \max ( 0, 100 \cdot v_x^{\top} c_y )$ when using the normalized
embeddings of the text ($v_x$) and image ($c_y$). While we have the max operator, unless $v_x^{\top} c_y$ is negative for all models, we can choose the best arm by $s' = v_x^{\top} c_y$. Moreover, even when each model $a$ has stochastic generation process of the image $y$, we can estimate the average score of $s$ as $\mathbb{E}[s|a] = v_x^{\top} (\mathbb{E}[c_y|a])$, where $(\mathbb{E}[c_y|a])$ can be represented by a linear vector. I could not understand why general Lin-UCB does not work in this setting.

- **Missing baselines in the experiments.**

The experiment section only compares the variants of the proposed algorithm, including kernel CB and random Fourier features (RFF). No other baselines, including Lin-UCB, is not compared, and it is not evident if the proposed method works better than the possible baselines.

- **Quality of writing.**

I realized some key components are not explained in the main text, making it hard to understand the proposed idea deeply. For instance, what is the difference between SCK-UCB-lin and SCK-UCB-poly? Also, what are the findings from the experiment section? It seems that the superiority among the compared methods changes with the experiment setting, but what affects the experiment results, and which algorithm we should use in practice (i.e., the experiment sections only report the results and lack the discussion)? The paper needs to address these questions in the main text. Also, the following are several small issues in the text, and I highly recommend restructuring and proofreading the paper once again.

[nits]

- Section 3.2; I guess $s$ refers to the evaluation score, but this variable is not formally defined.
- Remark 1; the sentence is not complete (end with "the")
- outcome-the-best (O2B); I think the definition is a bit ambiguous. I guess the "best individual model" refers to using a single model across the whole data, but I initially thought this was the best model chosen for individual inputs, i.e., I thought this was referring the regret at first.

**Questions:**

- How does the proposed method perform compared to Lin-UCB?

- What are the findings from the set of experiments?

- How are SCK-UCB-lin and SCK-UCB-poly different to each other?

- Empirically, how does the computation time differ among the methods?

---

> ### Author Response · Authors · 2024-11-22
> **Response to Reviewer WovK**
>
> We thank Reviewer WovK for his/her time and thoughtful feedback on our work. Here is our response to the reviewers' comments and questions.
>
> **1. Performance of Lin-UCB compared to general Kernel-UCB**
>
> If we correctly understood the reviewer's comment, Reviewer WovK suggests using a linear map to learn the relationship between the CLIP embedding $c_T$ of text $T$ and the CLIPSCore of the text-to-image model for the input prompt $T$. We believe this proposal will not generally address the task, and the choice of non-linear kernels is not just due to the max operator in the definition of CLIPScore.
>
> To understand the necessity of considering a non-linear model, note that the CLIPScore of a text-to-image model $G$ for text input $T$ with CLIP embedding $c_T$ is
> $$\text{CLIPScore}_G(T)= \max\bigl\\{ 0 , 100\cdot\langle c_T, G(T)\rangle\bigr\\}$$
> In this definition, the term $\langle c_T, G(T)\rangle$ is not linear in the CLIP embedding $c_T$, because the output of the text-to-image generator $G(T)$ depends on the text embedding $T$ in a potentially nonlinear way. In fact, due to the complexity of the text-to-image model $G$, one can expect the $\text{CLIPScore}_G(T)$ to be highly non-linear as a function of the CLIP embedding $c_T$.
>
> Consistent with the above intuition, we observed that the linear model in Lin-UCB could perform considerably worse than the non-linear model of a general Kernel-UCB method. Please note that the baseline named SCK-UCB-lin in our previously submitted draft indeed refers to the Lin-UCB. Our numerical results in Figures 2-7 on the comparison between Lin-UCB and SCK-UCB with nonlinear kernel functions all suggest that non-linear kernel-UCB can perform better than Lin-UCB.
>
>
> **2. SCK-UCB-poly3 and comparison to Lin-UCB**
>
> We would like to clarify that the SCK-UCB-lin algorithm (i.e., SCK-UCB with linear kernel given by $k_\text{lin}(x,y)=x^\top y$) in the experiments is the same as the Lin-UCB method. Additionally, the SCK-UCB-poly algorithm is SCK-UCB with the standard polynomial kernel of degree 3, i.e, $k_\text{poly3}(x,y)=(1+\gamma\cdot x^\top y)^3$. We have clarified the kernel definitions in the revision.
>
>
> **3. Typos in writing**
>
> We thank the reviewer for pointing out the typos in the paper.
> - Variable $s$ in Section 3.2: We note that the variable $s\in\mathbb{R}$ is the *target* for kernel ridge regression, which can be CLIPScore (or another performance measure) in the setting of prompt-based model selection.
> - Remark 1: We have revised Remark 1 in the updated draft.
> - O2B metric: To address the reviewer's comment, we have clarified in the revised text that O2B refers to the outscore with respect to the best single model.
>
>
> **4. Findings from the experiments**
>
> The main finding of our numerical experiments is the improvement of the proposed contextual bandit SCK-UCB algorithm over the one-arm oracle baseline that always chooses the model with the maximum averaged CLIPScore. This result means that the online learning algorithm can outperform a user with side-knowledge of the single best-performing model, which is made possible by a **prompt-based selection** of the model. This finding supports the application of contextual bandit algorithms in the selection of text-based generative models.
>
> Moreover, our numerical results indicate that the proposed SCK-UCB algorithm can perform better with a non-linear kernel function. Finally, in our experiments, the proposed RFF-UCB variant could reduce the computational costs of the general SCK-UCB algorithm. We have explained these findings more clearly in the revised text.
>
>
> **5. Computation time**
>
> To address the reviewer's question, we have compared the running time of RFF-UCB and SCK-UCB using the RBF kernel in Part 3 of Appendix A. The results show that RFF-UCB can achieve 5x acceleration compared to SCK-UCB with Gaussian RBF kernel.

---

> > ### Comment · Reviewer_WovK · 2024-11-22
> >
> > Thank you for the detailed explanations.
> >
> > **1. Performance of Lin-UCB compared to general Kernel-UCB**
> >
> > I understand that *the output of the text-to-image generator $G(T)$ depends on the text embedding $T$ in a potentially nonlinear way*. My point is that if the generated image ($y$) can be compressed to linear features $c_y$, it would be possible to learn the embeddings of $G(T)$ as $G(T) = \mathbb{E}[c_y|G, T]$, which is linear. This is the reason I was suggesting to estimate $G(T)$. But now it became clear that SCK-UCB-lin refers to Lin-UCB, and how it is difference to SCK-UCB-poly.
> >
> > ---
> >
> > Given the updated understanding of the paper, I will improve my score to 5. However, I remain on a rather negative size due to the following reasons.
> > - The idea of selecting different models depending on prompts can be interesting. This is a positive contribution.
> > - However, the technical contribution is weak as an ICLR paper, as the main contribution is speeding up the kernel computation of polynomial kernels (the idea of kernelized bandits has been long discussed by the community). Also, the speeding-up technique, RFF kernel, has also invented by an existing research paper.

---

> > > ### Author Response · Authors · 2024-11-25
> > > **Follow-up response to Reviewer WovK**
> > >
> > > We thank Reviewer WovK for his/her feedback on our response. Regarding the raised points in the feedback,
> > >
> > > **1. Idea of Learning the embedding of $G(T)$ from observed data**
> > >
> > > We thank the reviewer for the clarification. The reviewer's suggestion of learning the embedding of $G(T)$ also provides a contextual bandit algorithm to address the task. However, please note that we **only need** to estimate the CLIPScore in the online selection task, which is a 1-dimensional scalar. On the other hand, the CLIP embedding has a dimension of 512, for which learning the entire embedding vector could be more challenging due to the significantly larger dimension.
> > >
> > > **2. Problem formulation and algorithm design of our work vs. Lin-UCB and Kernelized UCB**
> > >
> > >
> > > We would like to clarify that the problem setting of our proposed SCK-UCB algorithm is **different** from the Lin-UCB [1] and Kernelized UCB [2] algorithms. In the following, we will compare the process in SCK-UCB and Kernelized UCB side-by-side to highlight their differences:
> > >
> > > - **Problem Setting of SCK-UCB (ours):** We have $N$ arms where each arm represents a **fixed generative model** that remains unchanged across rounds: for example, Arm 1 represents the Stable Diffusion model in all rounds. At each round, the arms observe **one shared** context variable (i.e., the text prompt). We learn $N$ **separate kernel-based models** with **different weights** to predict the CLIPScore of a shared incoming prompt (context) for the $N$ fixed generative models.
> > >
> > > - **Problem Setting of Lin-UCB [1] and Kernelized-UCB [2]:** At every round, we have $N$ arms where the expected reward of each arm is fully characterized by its context variable. The arms have **different** context variables at each round. We learn **one shared** set of weights to predict the expected reward for the $N$ observed contexts (i.e., arms) in the next round.
> > >
> > > As explained above, in the Lin-UCB [1] and Kernelized-UCB [2] settings, there is **not a fixed model** corresponding to one arm across iterations, and the arms will perform independently across iterations depending on their context. However, in the setting of SCK-UCB (our method), each arm will represent one fixed generative model in all the learning rounds. Therefore, we believe that our problem formulation and analysis are considerably different from the Lin-UCB [1] and Kernelized-UCB [2] methodologies.
> > >
> > > We hope that the above explanation helps with the reviewer's comment on the novelty of our problem formulation and algorithm design.
> > >
> > >
> > > [1] Chu, W., Li, L., Reyzin, L. and Schapire, R., 2011, June. Contextual bandits with linear payoff functions. In Proceedings of the Fourteenth International Conference on Artificial Intelligence and Statistics (pp. 208-214). JMLR Workshop and Conference Proceedings.
> > >
> > > [2] Michal Valko, Nathaniel Korda, Remi Munos, Ilias Flaounas, and Nelo Cristianini. Finite-time analysis of kernelised contextual bandits, 2013.

---

### Official Review · Reviewer_7U2V · 2024-11-02

**Soundness:** 3
**Presentation:** 3
**Contribution:** 2
**Rating:** 5
**Confidence:** 3

**Summary:**

In this paper, the authors work on selecting the optimal generative model for various text prompts, as different models may perform better on different types of prompts. They propose an online learning framework that utilizes a kernelized contextual bandit (CB) approach to dynamically predict the best generative model for each prompt. The proposed method, Shared-Context Kernel-UCB (SCK-UCB), updates a kernel-based function using observed prompt-model performance data to iteratively refine the model selection based on expected scores. To reduce computational demands, the authors introduce a variant with random Fourier features (RFF-UCB), which approximates SCK-UCB’s performance while lowering the computational complexity per iteration from cubic to linear time. The proposed framework is tested on tasks such as text-to-image generation and image captioning.

**Strengths:**

By adapting the kernelized contextual bandit framework for prompt-based selection, the authors introduce a method that dynamically identifies the best generative model according to prompt type, partially addressing the challenge of variable model performance across prompts. Further, the integration of random Fourier features (RFF) into the SCK-UCB algorithm significantly reduces computational complexity from cubic to linear time, enhancing the framework’s feasibility for real-world applications with constrained resources, all while maintaining high performance.

**Weaknesses:**

1. Although the RFF-UCB variant improves computational efficiency, the scalability of the proposed framework may still be limited in settings with a large number of prompts and models. The performance of SCK-UCB and RFF-UCB should be further explored under such conditions to determine their practical scalability in applications with larger datasets.

2. The manuscript briefly mentions selecting hyperparameters involved in the proposed method, such as the exploration parameter and kernel function. However, the authors might consider including a more detailed ablation study on hyperparameter sensitivity, which would provide insights. For instance, if the performance does not significantly depend on the hyperparameter, the robustness of the proposed method is then empirically supported. On the other hand, if the performance is sensitive to the hyperparameter, a detailed implementation of the hyperparameter selection would be beneficial.

**Questions:**

How does the framework handle cases where new generative models or prompt types are introduced after initial deployment? It remains unclear whether the proposed algorithms can adapt to or efficiently incorporate new options without retraining from scratch, which could be critical in certain applications.

---

> ### Author Response · Authors · 2024-11-22
> **Response to Reviewer 7U2V**
>
> We thank Reviewer 7U2V for his/her time and thoughtful feedback on our work. Here is our response to the reviewers' comments and questions.
>
> **1. Scalability of RFF-UCB**
>
> We believe that our proposed online learning approach can meet the scalability requirements of a typical generative AI user. In practice, users often have specific interests. For example, one user may mainly focus on sports and generate sports-related content. Our online learning approach is designed to efficiently adapt to these user-specific interests, by identifying the best generative model for prompts relevant to the user.
>
> For example, in the case of a user interested in sports, the algorithm adaptively learns the model assignment rules for sports-related prompts. This adaptation reduces computational and statistical costs by avoiding the need to find model assignments for all possible prompts, many of which would be irrelevant to the user. Therefore, by focusing on the user’s specific interests, our online learning method provides an efficient solution to the prompt-based model selection task.
>
> **2. Selection of hyperparameters**
>
> As requested by the reviewer, we have included the ablation study in Part 2 of Appendix A. We test 1) RFF-UCB with different selections on the hyperparameter in the RBF kernel function and the number of the random features, and 2) SCK-UCB-poly3 with different selections on the hyperparameter in the polynomial kernel and the regularization parameter in KRR. Our numerical results show that the proposed algorithms achieve consistent performance across different selections.
>
> **3. Adaptation to new models and prompt types**
>
> As requested by the reviewer, we included two numerical experiments in Section 6, where new prompt types and generators are introduced after initial deployment. The results show that the proposed SCK-UCB-poly3 algorithm works well in these scenarios.

---

### Official Review · Reviewer_T1Qe · 2024-11-04

**Soundness:** 2
**Presentation:** 3
**Contribution:** 2
**Rating:** 6
**Confidence:** 4

**Summary:**

This paper proposes to study a novel problem: generative model selection based on the given prompt and proposes a contextual-bandits-based algorithm to achieve sub-linear regret. Empirical results show that this method is effective in choosing the best model for each prompt.

**Strengths:**

1. The formulation of generative model selection based on a given prompt using contextual bandits is clear to me. The choose of bandits algorithm is well motivated.

 2. The presentation is clear to me.

**Weaknesses:**

1. This paper did not include enough discussion on the existing works that use bandits algorithm to solve the selection of prompts/models. Two examples are [1,2]. More discussion on this is needed to position this work.

2. The experimental results seem to be weak since the author only compare with the method proposed in this paper not use any other existing methods. An intuitive approach will be random selection (i.e., selecting the models randomly in the proposed algorithm, instead of using UCB).

3. The theoretical results seem to be standard. The regret of kenerlized bandits/contextual bandits has already shown in many bandits works and the use of RFF to approximate the kernel regression process is also standard since it is heavily used in previous work. My recommendation is that if there are novelty in the proof, please specify. If not, I think the theories in main paper seem to be redundant and can be removed to focus more on empirical insights.





[1] Chen, L., Chen, J., Goldstein, T., Huang, H., & Zhou, T. (2023). Instructzero: Efficient instruction optimization for black-box large language models. ICML 2024.
[2] Lin, X., Wu, Z., Dai, Z., Hu, W., Shu, Y., Ng, S. K., ... & Low, B. K. H. (2023). Use your instinct: Instruction optimization using neural bandits coupled with transformers. ICML 2024.

**Questions:**

see weakness

---

> ### Author Response · Authors · 2024-11-22
> **Response to Reviewer T1Qe**
>
> We thank Reviewer T1Qe for his/her time and thoughtful feedback on our work. Here is our response to the reviewers' comments and questions.
>
> **1. Discussion on related work**
>
> We thank the reviewer for pointing out the related works. We note that the mentioned references concern the *online training* of generative models, which is a different task from ours on the *online selection* among a group of already-trained generation models to reach the best performance in response to an input prompt. We have discussed these references in the revised draft.
>
> **2. Additional baselines**
>
> As requested by the reviewer, we included two additional baselines in our testing of the proposed kernel UCB algorithm, which are included in the plots of our revised draft. The baselines include: 1) the reviewer's suggested pure random selection of the model, and 2) the one-arm oracle baseline where the user has side-knowledge of the averaged CLIPScore of each generative model and always picks the model with the maximum averaged CLIPScore. Our numerical results in the revised draft indicate that the proposed SCK-UCB can perform better than both these baselines.
>
> **3. Novelty in the technical contribution**
>
> First, we would like to clarify that our proposed SCK-UCB algorithm is not a special case of the kernelized UCB approach in [1]. Note that the kernelized UCB [1] assumes a shared weight for the model's score function to an input and the context variable is different across arms. In contrast, in our setting, the score functions are different between arms, as the generative models can perform differently in response to the same prompt. Also, the context variable (text prompt) is shared between models in our formulation, which is also different from the kernelized UCB [1].
>
> Since the problem formulation in our case and the existing kernelized UCB literature are different, none of our theoretical regret bounds will follow from existing results in the literature. In other words, the formulated problem setting, proposed algorithms SCK-UCB and RFF-UCB, and the regret bounds are all novel results which, to the best of our knowledge, have not been discussed in any existing work.
>
> **4. Novelty in the Proofs**
>
> To highlight the novelties in the paper's proofs,
>
> - The existing regret analyses in the literature do not apply to random Fourier features for approximating a shift-invariant kernel in the kernel-UCB setting. Our regret analysis in Theorem 2 addresses this task.
> - The existing results cannot be applied to derive finite-sample error bounds on both the estimated mean score $\widetilde{\mu}_g$ and the uncertain quantifier $\widetilde{\sigma}_g$ (lines 8 and 9 in Algorithm 3). We establish the error bounds in Lemmas 5 and 6 in Appendix C.1.
>
> [1] Michal Valko, Nathaniel Korda, Remi Munos, Ilias Flaounas, and Nelo Cristianini. Finite-time analysis of kernelised contextual bandits, 2013.

---

> > ### Comment · Reviewer_T1Qe · 2024-12-02
> >
> > Thank you for your additional results which indeed addressed my concern on the comparison. I have increase my score.

---

> > > ### Author Response · Authors · 2024-12-03
> > >
> > > We thank Reviewer T1Qe for the time and feedback on our response. We are glad to hear that our response has addressed the reviewer's concern.

---

### Author Response · Authors · 2024-11-22

We thank the reviewers for their thoughtful feedback and suggestions. We have responded to each reviewer's comments and questions under the review textbox. We have uploaded the revised paper, which includes additional numerical results for 1) random selection baseline as requested by Reviewer T1Qe, 2) ablation study on hyperparameters, and 3) experiments with new prompts and models as requested by Reviewer 7U2V. Also, we have reorganized Section 6 to improve readability and have relocated part of the numerical results to the Appendix to create enough space for the new results in the main text.

---

### Meta-Review · Area_Chair_EbWV · 2024-12-21

**Metareview:**

(a) Summary of Scientific Claims and Findings

This paper introduces an online learning framework for dynamically selecting the most suitable generative model for a given input prompt. The work is driven by the observation that different generative models excel with specific prompts, and a dynamic selection strategy can effectively minimize the cost of querying suboptimal models.


(b) Strengths of the Paper

The authors employ a contextual bandit framework where the input prompt serves as a shared context across all arms (representing different models).
To enhance computational efficiency, they propose the RFF-UCB variant, which uses random Fourier features (RFF) to approximate the kernel, reducing computational complexity from cubic to linear per iteration.
Additionally, the paper establishes a regret bound for the RFF-UCB algorithm, guaranteeing sub-linear regret over multiple iterations.


(c) Weaknesses of the Paper and Missing Elements

While the framework builds on existing methodologies (e.g., kernelized UCB), its primary novelty lies in the application and integration of RFF for scalability.
The problem formulation could benefit from stronger justification, and the experimental results rely heavily on synthetic environments, limiting practical applicability.

(d) Decision and Rationale

This paper offers an intriguing contextual bandit approach to the problem of generative model selection. However, the theoretical contributions and experimental evaluation require further refinement. Highlighting theoretical innovations, particularly proof steps tailored to the problem setting, and providing more realistic experimental scenarios would significantly improve the work.

**Additional Comments On Reviewer Discussion:**

The authors addressed many of the reviewers’ concerns during discussions. However, some skepticism remains regarding the novelty of the theoretical contributions.

---

### Decision · Program_Chairs · 2025-01-22

Reject